



# Adaptation of root zone storage capacity to climate change and its effects on future streamflow in Alpine catchments: towards non-stationary model parameters

Magali Ponds[1,2], Sarah Hanus[3], Harry Zekollari[2,4,5], Marie-Claire ten Veldhuis[1], Gerrit Schoups[1], Roland Kaitna[6], and Markus Hrachowitz[1]

[1]Faculty of Civil Engineering and Geosciences, Delft University of Technology, Delft, the Netherlands
[2]Department of Water and Climate, Vrije Universiteit Brussel, Brussel, Belgium
[3]Department of Geography, University of Zurich, Zürich, Switzerland
[4]Laboratory of Hydraulics, Hydrology and Glaciology (VAW), ETH Zürich, Birmensdorf, Switzerland
[5]Laboratoire de Glaciologie, Université libre de Bruxelles, Brussels, Belgium
[6]University of Natural Resources and Life Sciences Vienna, 1180 Wien, Austria

**Correspondence:** Magali Ponds (magali.ponds@vub.be)

**Abstract.** Hydrological models play a vital role in projecting future changes in streamflow. Despite the strong awareness of non-stationarity in hydrological system characteristics, model parameters are typically assumed to be stationary and derived through calibration on past conditions. Integrating the dynamics of system change in hydrological models remains challenging due to uncertainties related to future changes in climate and ecosystems.

Nevertheless, there is increasing evidence that vegetation adjusts its root zone storage capacity – considered a critical parameter in hydrological models – to prevailing hydroclimatic conditions. This adaptation of the root zone to moisture deficits can be estimated by the Memory Method. When combined with long-term water budget estimates in the Budyko framework, the Memory method offers a promising approach to estimate future climate-vegetation interaction and thus time-variable parameters in process-based hydrological models.

Our study provides an exploratory analysis of non-stationary parameters for root zone storage capacity in hydrological models for projecting streamflow in six catchments in the Austrian Alps, specifically investigating how future changes in root zone storage impact modeled streamflow. Using the Memory method, we derive climate-based parameter estimates of the root zone storage capacity under historical and projected future climate conditions. These climate-based estimates are then implemented in our hydrological model to assess the resultant impact on modeled past and future streamflow.

Our findings indicate that climate-based parameter estimations significantly narrow the parameter ranges linked to root zone storage capacity. This contrasts with the broader ranges obtained solely through calibration. Moreover, using projections from 14 climate models, our findings indicate a substantial increase in the root zone storage capacity parameters across all catchments in the future, ranging from +10% to +100%. Despite these alterations, the model performance remains relatively consistent when evaluating past streamflow, independent of using calibrated or climate-based estimations for the root zone storage capacity parameter. Additionally, no significant differences are found when modeling future

streamflow when including future climate-induced adaptation of the root zone storage capacity in the hydrological





model. Variations in annual mean, maximum, and minimum flows remain within a 5% range, with slight increases found for monthly streamflow and runoff coefficients. Our research shows that although climate-induced changes in root zone storage capacity occur, they do not notably affect future streamflow projections in the Alpine catchments under study. Our findings suggest that incorporating a dynamic representation of the root zone storage capacity parameter may not be crucial for modeling streamflow in humid and energy-limited catchments. However, our observations indicate relatively larger changes in root zone storage capacity within the less humid catchments, corresponding to higher variations in modeled future streamflow. This suggests a potentially higher importance of dynamic representations of root zone characteristics in arid regions and underscores the necessity for further research on non-stationarity in these regions.

## 1 Introduction

Climate change is expected to further increase global temperature and precipitation extremes in the future, thereby causing the hydrological cycle to accelerate (IPCC, 2023). In combination with direct land-use change by humans (e.g. deforestation), climate change affects vegetation and its crucial role in the terrestrial water cycle through changes in overall plant biomass, species distribution and water use efficiency (Stephens et al., 2021). While difficult to generalize, there has been recent progress in quantitatively describing the effect of deforestation and reforestation on the hydrological response with time-variable parametrizations of hydrological models (e.g. Zhang et al., 2017; Teuling et al., 2019;Nijzink et al., 2016; Hrachowitz et al., 2021). In contrast, the natural adaptation of ecosystems to a changing climate is less well understood, due to the complex feedbacks between soils, vegetation and climate (Stephens et al., 2021) and gradual long-term changes, as opposed to rather abrupt human-induced land use change (Seibert and van Meerveld, 2016). These feedbacks complicate reliably predicting the future hydrological responses of catchments under change, which is recognized as a major challenge in hydrology (Blöschl et al., 2019; Berghuijs et al., 2020).

The quantification of how physical characteristics of a terrestrial hydrological system are affected by climate change is complex. In the absence of that knowledge, model parameters, observed or calibrated to past observations, are typically used for predicting future hydrological responses. However, inferring model parameters from historic conditions, requires the implicit assumption that the considered system is stationary and that its physical characteristics (and thus model parameters), such as vegetation root systems, do not change over the modeling period. Although this assumption may hold for predictions on shorter timescales, assuming long-term system stationarity under a changing climate may lead to misrepresentations of the underlying processes and results in considerable associated predictive uncertainties (e.g. Fenicia et al., 2009; Hrachowitz et al., 2021; Bouaziz et al., 2022).

There is increasing evidence that vegetation dynamically adapts its root systems to the prevailing climate in order to guarantee water supply to satisfy the canopy water demand by transpiration (Kleidon, 2004; Schymanski et al., 2009). As such, changes in the root system change the soil pore volume and thus the water volume between permanent wilting point and field capacity that is within reach and can be accessed by vegetation roots to satisfy water demand in dry periods. This maximum vegetation-accessible water volume is hereafter referred to as the 'root zone storage capacity'. Regulating the





water supply for vegetation, this root zone storage capacity plays a key role in the partitioning of water fluxes in terrestrial hydrological systems, where it regulates the temporally variable ratio between drainage and evaporative water fluxes (Rodriguez-Iturbe et al., 2007; Savenije and Hrachowitz, 2017; Gao et al., 2024; Van Oorschot et al., 2021). Consequently, changes in root systems are also reflected in changes in transpiration and streamflow (Gao et al., 2014; Zhang et al., 2001; **?**). This makes the root zone storage capacity ($S_r$) a core parameter in hydrological models, where its value is typically

inferred from (i) observations (e.g. from soil characteristics and estimates of root depth) or (ii) through calibration (Andréassian et al., 2003). However, detailed observations of root depths are scarce in both space and time and are difficult to extrapolate to the catchment scale, due landscape and the vegetation cover heterogeneity (Wagener, 2007; Duethmann et al., 2020). Both the observations and the calibration do merely provide windows into the past, as the estimated values of $S_r$ are a result of the adaptation of vegetation to past climatic conditions. Consequently, the use of $S_r$ estimated from past

climatic conditions for model predictions in a changing climate may lead to a misrepresentation of the water constraint for transpiration (Jiao et al., 2021) and thus to considerable uncertainties. An explicit representation of vegetation responses to changing climate conditions expressed through temporally variable (i.e. non-stationary) model parameters may therefore prove a valuable step towards more reliable predictions (Coron et al., 2012; Keenan et al., 2013).

Optimality principles, that consider the co-evolution of climate, soil and vegetation in a holistic way (Blöschl, 2010), may

offer an alternative to quantify a temporal variability in $S_r$ and root systems changes (e.g. Kleidon, 2004; Gentine et al., 2012; Gao et al., 2014; de Boer-Euser et al., 2016; Wang-Erlandsson et al., 2016; Hrachowitz et al., 2021; McCormick et al., 2021; Terrer et al., 2021; Stocker et al., 2023). Data from a wide range of contrasting environments support the hypothesis that, for example, forests invest just enough resources to develop root systems large enough to guarantee sufficient access to water (and nutrients) during droughts with around 20-40 years return period, but, importantly, not more than that

to ensure an efficient distribution of energy and resources between below-surface and above-surface growth (e.g. Guswa, 2008). The presence of vegetation at any location at any time implies that this vegetation has had sufficient access to water to satisfy canopy water demand by transpiration to survive past dry periods. By extension, at any point in time, the maximum storage deficit $S_{r,D}$ between precipitation $P$ and transpiration $E_R$, occurring over the previous 20-40 years, is then a robust first order estimate of the available $S_r$ for that period, since this is the necessary storage to sustain the observed

transpiration over the driest year (cf. Hrachowitz et al., 2021). This implies that storage deficits, and thus the root zone storage capacity $S_r$, can be estimated exclusively based on water balance data, i.e. time series of $P$ and $E_R$. Changes in hydroclimatic conditions will therefore manifest in time-variable estimates $S_R$ over medium- to long time scales, reflecting the adaptation of the vegetation transpiration response to these hydroclimatic changes (Tempel et al., 2024; Wang et al., 2024).

Following the above reasoning, future projections of changes in hydroclimatic variables may then, under certain assumptions, also allow for first-order estimates on how the root zone storage capacity $S_R$ changes over time. Such projections of future precipitation ($P$) and temperature, as proxy of atmospheric water demand (or potential evaporation, $E_P$), are readily available from climate models. In contrast, many studies underline future transpiration ($E_R$) estimates are subject to more pronounced uncertainties (Milly and Dunne, 2011; Wartenburger et al., 2018), partly related to the use of time-





invariant representation of $S_R$ in the vast majority of current climate models (Van Oorschot et al., 2021). To avoid the need for climate-model-derived estimates of $E_R$ and the apparent circular argument arising from using time-invariant values of $S_R$ in those models, the Budyko hypothesis provides an interesting alternative. Following this hypothesis, the long-term hydroclimatic conditions, expressed as the aridity index $AI = E_P/P$, are a dominant control on the water budget and thus on long-term average $E_R$ of a catchment. Notwithstanding uncertainties and additional effects arising from the co-evolution

of landscape and vegetation properties with climate characteristics over time (e.g. Zhang et al., 2004; Troch et al., 2013; Jaramillo et al., 2018; Berghuijs et al., 2020; Ibrahim et al., 2024), future projections of $P$ and $E_P$ thus allow for first-order estimates of the associated future $E_R$.

Hence, combining the above-described Memory method to estimate root zone storage capacities $S_R$ with projections of long-term future water budget estimates based on the Budyko hypothesis, provides a step towards quantifying how climate

change influences hydrological system characteristics and parameters, and how these, in turn, affect the future hydrological response (Zhang et al., 2001; Bouaziz et al., 2022).

This study builds on the work by Hanus et al. (2021), who investigated future streamflow in alpine catchments at varying elevations, given that alpine regions are expected to experience significant impacts from climate change. Consequently, all results regarding future streamflow projections using stationary model parameters were already presented in Hanus

et al. (2021). In this study, we extend the analysis of future streamflow for the same catchments and using the same climate model data to quantify the potential *additional* effects of a time-variable formulation of the root zone storage parameter $S_R$ in a process-based hydrological model. Utilizing the Memory method to estimate time-variable values of $S_R$ (Bouaziz et al., 2022), we test how a time-variable formulation of the root zone storage capacity parameter $S_R$ impacts the projected hydrological response pattern for the period 2070-2100.


## 2   Study Area & Data

### 2.1   Study Area

The study examines six contrasting catchments in the Austrian Alps, which cover a spectrum of hydro-climatic regimes and landscape types (Figure 1, Table 1). The dominant land cover at high elevations is bare rock and grassland, whereas

lower elevation catchments are mainly covered by forest.

The Pitztal has the highest mean elevation (2558m) and features a nivo-glacial discharge regime. This catchment is located in the west of Austria and is, due to its elevation, characterized by a large fraction of sparsely vegetated soils (70%, with 18% glacial coverage). The lowest-elevation catchment is the Feistritztal, with a mean elevation of 917m. The Feistritztal is located in the east of Austria, featuring a nivo-pluvial discharge regime and a relatively dense vegetation cover (72% forest

and 25% grass). All other catchments, with mean elevations between 1315m (Paltental) and 2233m (Defreggental), exhibit a nival regime. The land cover varies in correspondence with elevation:



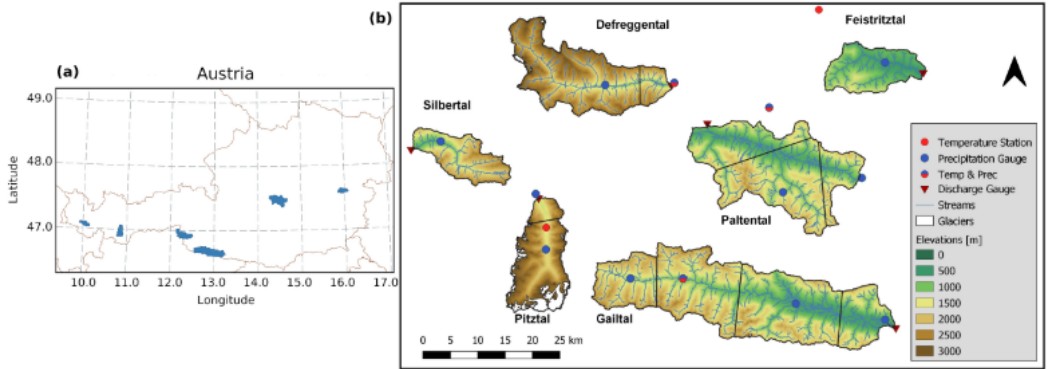

**Figure 1. (a)** Location of the catchments in Austria. **(b)** Catchment outlines indicating altitude, different precipitation zones and the location and number of Precipitation Gauges and Temperature Stations (Right) (Hanus et al., 2021)

**Table 1.** Catchment characteristics, land cover data & discharge regimes based on Mader et al. (1996) and historical climate data (1985-2005)

|  | Feistritztal | Gailtal | Paltental | Silbertal | Defreggental | Pitztal |
|---|---|---|---|---|---|---|
| **Characteristics** |  |  |  |  |  |  |
| Mean Altitude [m] | 917 | 1476 | 1315 | 1776 | 2233 | 2558 |
| Elevation [m] | 449-1595 | 596-2778 | 633-2447 | 671-2764 | 1096-3763 | 1339-3763 |
| Area [km$^2$] | 116 | 587 | 370 | 100 | 267 | 166 |
| Prec. Gauges [#] | 1 | 4 | 3 | 1 | 2 | 2 |
| Discharge Regime | Nivo-pluvial | Autumn nival | Moderate nival | Nival | Nival | Nivo-glacial |
|  |  |  |  |  |  |  |
| **Land cover** |  |  |  |  |  |  |
| Bare (Glacier) [%] | 0 (0) | 8 (0) | 4 (0) | 20 (0) | 43 (1.5) | 70 (18) |
| Grass [%] | 25 | 33 | 32 | 46 | 32 | 23 |
| Forest [%] | 72 | 57 | 61 | 32 | 23 | 6 |
| Riparian [%] | 3 | 2 | 3 | 2 | 2 | 1 |
|  |  |  |  |  |  |  |
| **Climate** |  |  |  |  |  |  |
| Temperature [°C] | 9.28 | 5.23 | 7.85 | 8.99 | 4.16 | 5.14 |
| Potential Evaporation [mm d−1] | 1.59 | 1.30 | 1.39 | 1.30 | 1.05 | 1.03 |
| Discharge [m$^3$/s] | 0.95 | 2.53 | 2.18 | 2.76 | 1.75 | 1.47 |
| Precipitation [mm d−1] | 2.30 | 3.58 | 3.29 | 3.95 | 2.50 | 2.54 |





**Table 2.** EURO-CORDEX projections used in this study

| ID | GCM | RCM | ID | GCM | RCM |
|----|-----|-----|----|-----|-----|
| 1 | CNRM-CM5 r1i1p1 | CCLM4-8-17 | 8 | CM5A-MR r1i1p1 | WRF361H |
| 2 | CNRM-CM5 r1i1p1 | ALADIN53 | 9 | CM5A-MR r1i1p1 | RCA4 |
| 3 | CNRM-CM5 r1i1p1 | RCA4 | 10 | HadGEM2-ES r1i1p1 | CCLM4-8-17 |
| 4 | EC-EARTH r1i1p1 | RACMO22E | 11 | HadGEM2-ES r1i1p1 | RCA4 |
| 5 | EC-EARTH r3i1p1 | HIRHAM5 | 12 | HadGEM2-ES r1i1p1 | RACMO22E |
| 6 | EC-EARTH r12i1p1 | CCLM4-8-17 | 13 | MPI-ESM-LR r1i1p1 | CCLM4-8-17 |
| 7 | EC-EARTH r12i1p1 | RCA4 | 14 | MPI-ESM-LR r1i1p1 | RCA4 |

## 2.2 Data

This study provides a past-future analysis of modelled streamflow covering a period of 30 years:1981-2010 and 2071-2100. The deployed datasets are elaborated upon below.


### 2.2.1 Observational data (1981-2010)

Topographic information is derived from a 10 × 10m digital elevation model (DEM) of Austria (https://www.data.gv.at /katalog/dataset/dgm) and land cover data from the CORINE Land Cover dataset ( https://land.copernicus.eu/pan- e uropean-corine-land-cover, 2018) (Table 1). Historic glacier outlines between 1997-2006 are available from the Austrian

Glacier Inventory (https://www.uibk.ac.at/en/acinn/research/ice-and-climate/projects/austrian-glacier-inventory/) (Lambrecht and Kuhn, 2007; Abermann et al., 2010). Glacial area changes are determined through linear interpolation on the observed outlines from 1997 to 2006, and subsequently extrapolated to estimate glacier areas up to 2015. Additionally, future glacier extents under various emission scenarios are available for the Pitztal from Zekollari et al. (2019), who simu- lated the future evolution of European glaciers using GloGEMflow. This model is an enhanced version of the Global Glacier

Evolution Model by Huss and Hock, 2015, that explicitly considers ice flow. The resulting future glacier extents under dif- ferent emission scenarios are used in this study and are scaled to match extrapolated glacier areas in 2015.

### 2.2.2 Projected data (1981-2010 and 2071-2100)

For a meaningful comparison between past and future hydrological responses, we rely on climate model projections of
precipitation and temperature. These projections are derived at the station scale for a historical (1981-2010) and future (2071-2100) period from 14 high-resolution regional climate models within the EURO-CORDEX ensemble (Table 2). Pre- cipitation and temperature data are provided on a daily basis at the station scale corresponding to the locations of precip- itation and temperature stations (Figure 1). Bias correction is applied using scaled distribution mapping, with a gamma





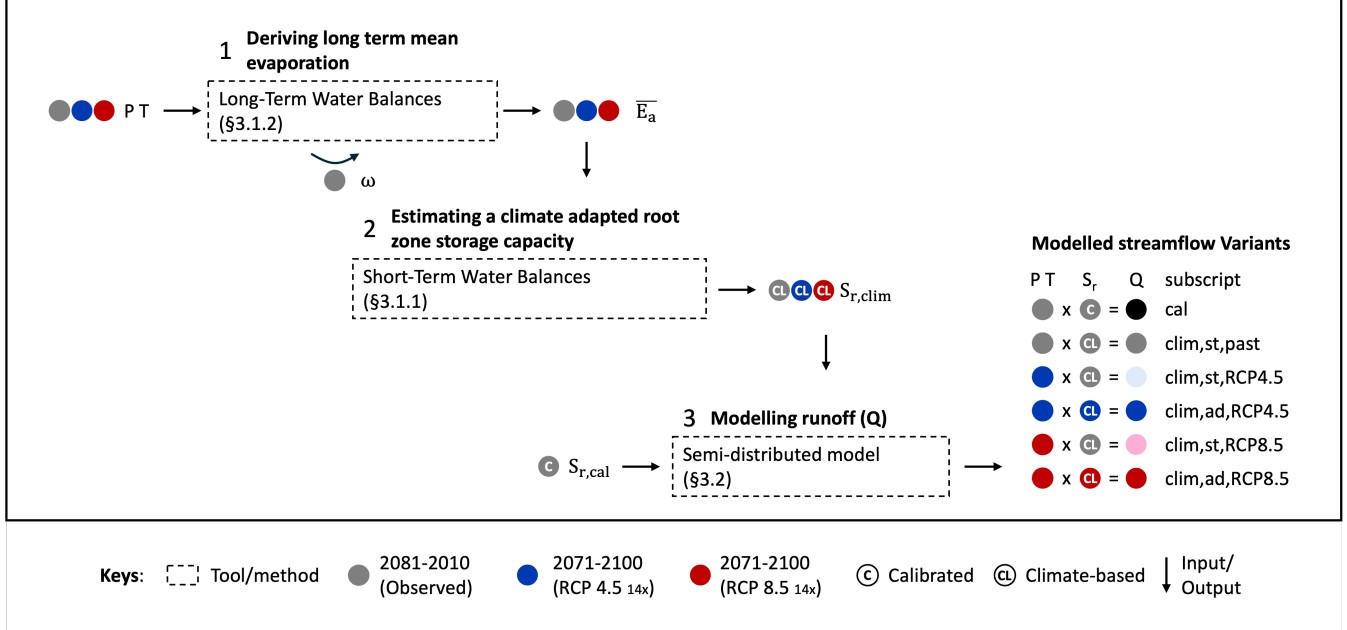

**Fig. 2.** Process scheme of step wise approach. The steps followed are: Estimate long term runoff coefficient from the Budyko framework (3.1.2). Determine climate-based root zone storage capacity values (3.1.1). Implement calibrated $S_{r,cal}$ as well as climate-based $S_{r,clim,past}$ and $S_{r,clim,fut}$ to model past and future streamflow, respectively (3.2). Numbering refers to the associated Method sections. Abbreviations used include $E_a$ for actual evaporation, $P$ for precipitation, $T$ for temperature, $S_{r,cal}$ for calibrated root zone storage capacity parameter, $S_{r,clim}$ for climate-based root zone storage capacity parameter. Subscripts $_{ad}$ denote parameters adapted to future climate, while $_{st}$ signifies parameters in stationary conditions.

distribution to remove systematic model errors (Switanek et al., 2017). For each regional climate model, we consider the
two emission scenarios RCP4.5 and RCP8.5. RCP4.5 represents an intermediate pathway with partially reduced emissions, resulting in a radiative forcing of 4.5Wm$^{-2}$ by 2100. In contrast, RCP 8.5 represents a trajectory characterized by increasing greenhouse gas emissions without mitigation measures.

## 3 Methods

This study adopts a top-down and process-based approach, with the main aim to describe the impact of climate change on
vegetation (Section 3.1) and the impacts thereof on streamflow in the past and future (Section 3.2). The study follows a five-step procedure: First, (1) the root zone storage capacity model parameters $S_{r,clim,past}$ for the six study catchments are estimated from past water balance data (Section 3.1.1), then (2) future climate projections are combined with long-term water budget estimates following the Budyko hypothesis (Section 3.1.2) to estimate the corresponding future root zone storage capacities $S_{r,clim,fut}$ (Section 3.1.3). Using a semi-distributed, process-based hydrological model, we then (3) compare





$S_{r,clim,past}$ to the corresponding values $S_{r,cal}$ obtained by model calibration on past data and the associated model performances. Subsequently, we (4) apply $S_{r,clim,past}$ together with future time series of $P$ and $T$ for RCP4.5 and RCP 8.5 from an ensemble of 14 climate projections to quantify the effect of a changing climate on the hydrological response assuming stationary catchment properties and thus time-invariant parameters (Section 3.2.2) and, on the condition that the model using the climate-based $S_{r,clim,past}$ can reasonably describe past streamflow signatures, (5) apply $S_{r,clim,fut}$ with future climate projections to quantify the additional effects of non-stationary and thus time-variant root zone storage capacities on the hydrological response (Section 3.2.3).

## 3.1 Describing time-variant climate vegetation interactions

### 3.1.1 Using short-term water balances to quantify vegetation root-zone storage capacity

Previous studies have shown that different vegetation types develop root systems that can bridge droughts of varying return periods (Wang-Erlandsson et al., 2016). Following this assumption, the root zone storage capacity of riparian vegetation, grass and forest develops to endure droughts with a return period of respectively 2, 2 and 20 years. The return period for riparian vegetation has been approximated to be equal to that of grass. Corresponding root zone storage capacities for grass ($S_{r,clim,grass}$) and forest ($S_{r,clim,forest}$) can then be estimated from the time-series of maximum annual storage deficits $S_{r,D,yr}$, using the Gumbel extreme value (GEV) distribution (e.g. de Boer-Euser et al., 2016; Nijzink et al., 2016).

The Memory method builds on this principle, describing how vegetation root zones adapt to prevailing climate conditions by creating a water buffer within reach of the roots, sufficient to bridge dry spells (Gentine et al., 2012; Donohue et al., 2012; Gao et al., 2014). We employ the Memory method to estimate the annual maximum excess of transpiration $E_R$ over effective precipitation $P_E$. As a first approximation, this excess transpiration is assumed to originate from the water stored in the unsaturated zone. However, for various landscape types and vegetation species roots may also directly tap groundwater (e.g. Fan et al., 2017). As input for the Memory method, we use daily time series of effective precipitation $P_E$ and transpiration $E_R$. Effective precipitation, defined as the liquid water input from snow-melt ($M$) plus rainfall after interception ($I = -\frac{dS_I(t)}{dt} - E_I(t)$), is estimated from the water balance of the canopy storage (Equation 1). As the interception storage capacity ($I_{max}$) of the interception storage ($S_I$) is unknown, a random sample of 300 a-priori constrained $I_{max}$ values is used (Supplement S2).

$$P_E(t) = P(t) + M(t) - E_I(t) - \frac{dS_I(t)}{dt} \tag{1}$$

The long-term mean transpiration ($\overline{E}_R$) is approximated by the long-term evaporation ($\overline{E}_A$) and derived from the long-term mean water balance (Equation 2, all in mm yr$^{-1}$). This approximation operates under the assumption that long-term inter-catchment groundwater exchange and storage changes are negligible. The long-term mean transpiration is subsequently scaled to daily transpiration estimates, using the daily difference between potential evaporation $E_P$ and intercep-





tion evaporation $E_I$ (Equation 3). Here, potential evaporation is estimated using the Thornthwaite equation (Yates, 1994, Li et al., 2018). Note that the transpiration estimate also includes soil evaporation, as these fluxes can not be separated from the available data. Nonetheless, it has been demonstrated that transpiration significantly exceeds soil evaporation fluxes
(e.g. Jasechko et al., 2013).

$$\overline{E}_R \approx \overline{P}_E - \overline{Q} \tag{2}$$

$$E_R(t) = (E_P(t) - E_I(t)) \cdot \frac{\overline{E}_R}{(\overline{E}_P - \overline{E}_I)} \tag{3}$$

Starting from daily deficits, we estimate the total water buffer stored in the root zone by accumulating over the total period of water shortage $(T_0 - T_1)$ (Equation 4). Thereby, $T_0$ marks the first day at which transpiration exceeds effective
precipitation $((P_E - E_R) < 0)$ and $T_1$ represents the day at which the root zone storage deficit is restored to zero $(S_{r,D} = 0)$. The yearly maximum storage deficits $S_{r,D,yr}$ (Equation 5) is then selected and used as input for the GEV distribution to calculate $S_r, clim$ for grass, forest and riparian vegetation. Note that since the derived estimates of $S_{r,clim,fut}$ values represent root zone storages in catchments that are entirely covered by either forest, grass or riparian vegetation, these estimates may over- or underestimate the actual root zone storage capacity of a given catchment. Hence the derived $S_{r,clim}$ values are
multiplied with the respective areal share of each vegetation type.

$$S_{r,D}(t) = \min\Big(0, S_{r,D}(t-1) + \big(P_E(t) - E_R(t)\big)\Big) \tag{4}$$

$$S_{r,D,yr} = \min_{yr} \sum_{T_0 \le t \le T_1} S_{r,D}(t) \tag{5}$$

While data for the historic time period is readily available, the long-term mean runoff for the future remains unknown and can be derived through the application of the Budyko framework.


### 3.1.2 Long term water balance framework for estimating changes in future runoff

To estimate future evaporation under a changing climate, we here use (i) time-series of projected future $P$ together with (ii) estimates of future $E_P$ based on projected $T$, combined with (iii) the long-term water-balance, as described by the Budyko hypothesis (e.g. Turc, 1954; Mezentsev, 1955; Budyko, 1961; Fu, 1981; Zhang et al., 2004). This hypothesis describes how
climate - expressed as the aridity index $(\frac{\overline{E}_P}{\overline{P}})$ controls the long-term partitioning of precipitation $(\overline{P})$ into evaporation $(\overline{E}_A)$ and streamflow $(\overline{Q})$. The Budyko space is bound by (i) the supply limit, as no more water can evaporate than is available, and (ii) the demand limit, with evaporation not being able to exceed the potential evaporation (Zhang et al., 2001; Xing





et al., 2018; Mianabadi et al., 2020; Berghuijs et al., 2020). Despite its relatively simple structure and low requirement for input data, the Budyko curve broadly captures the partitioning of water fluxes in virtually every catchment around the
world (Berghuijs et al., 2020).

However, the original Budyko relationship does not explicitly consider the combined influence of soil, topography and vegetation, possibly explaining the systemic scatter found around the Budyko curve (Troch et al., 2013). As an attempt to overcome this limitation, Budyko equations, such as the Fu equation, account for bulk catchment biophysical features through the catchment-specific parameter ($\omega$) (Equation 6; Tixeront, 1964; (Fu, 1981).

$$\frac{\overline{E_A}}{\overline{P}} = 1 - \frac{\overline{Q}}{\overline{P}} = 1 + \frac{\overline{E_P}}{\overline{P}} - \left(1 + \left(\frac{\overline{E_P}}{\overline{P}}\right)^{\omega}\right)^{\frac{1}{\omega}} \tag{6}$$

Despite ongoing attempts (Dwarakish and Ganasri, 2015; Sankarasubramanian et al., 2020; Jaramillo et al., 2018; Jaramillo and Destouni, 2014; Jaramillo et al., 2018; Van der Velde et al., 2014), the heterogeneity and interdependency of catchment-specific influences make it difficult to meaningfully disentangle the role of individual influencing factors for $\omega$. Therefore, the relationship between evolving vegetation dynamics and changes in the catchment-specific parameter needs to be as-
sumed to be specific at the catchment scale. Here, the value of $\omega$ is estimated by solving Equation 6, using observed climate and streamflow data, averaged over the historical 30-year study period (1981-2010), ($P_{obs}$, $T_{obs}$, $Q_{obs}$). The resulting parameter value ($\omega_{obs}$) hence reflects historical catchment conditions.

We here assume the general pattern of water partitioning in the Budyko hypothesis, describing a past dynamic equilibrium, will not drastically change under future conditions (Jaramillo et al., 2022). This simplification allows us to work
towards a time-variant estimate of $S_{r,clim}$ and is justified by uncertainties regarding the magnitude of effects stemming from alterations in vegetation cover and adjustments in vegetation water use efficiency in response to fluctuating atmospheric CO2 levels (e.g. Yang et al., 2021). Under this assumption, the $\omega_{obs}$-parameterized Budyko curve derived from past climate conditions can be used to broadly estimate changes in the future partitioning of water fluxes.

More specifically, future changes in climate are reflected in a shift of the aridity index AI, as a consequence of changes in
precipitation rates ($\Delta\overline{P} = \overline{P}_{fut} - \overline{P}_{obs}$), temperature and hence potential evapotranspiration ($\Delta\overline{E}_P = \overline{E}_{P,fut} - \overline{E}_{P,obs}$) (Equation 7; Figure 3). These changes in climate cause a horizontal shift in the Budyko space, moving a catchment from its initial position ($p_{obs}$), along the $\omega_{obs}$-parameterized Budyko curve, to a new position ($p_{fut}$). From the new long-term average future evaporation and streamflow can be inferred from the evaporative index ($E_{A,fut}/P_{fut}$) and runoff ratio ($\frac{Q_{fut}}{P_{fut}} = 1 - \frac{E_{A,fut}}{P_{fut}}$), respectively (Equation 8).

$$\left(\frac{\overline{E_P}}{\overline{P}}\right)_{fut} = \frac{\overline{E}_{P,obs} + \Delta\overline{E}_P}{\overline{P}_{obs} + \Delta\overline{P}} \tag{7}$$

$$\left(\frac{\overline{Q}}{\overline{P}}\right)_{fut} = \frac{\overline{Q}_{obs} + \Delta\overline{Q}}{\overline{P}_{obs} + \Delta\overline{P}} = -\left(\frac{\overline{E_P}}{\overline{P}}\right)_{fut} + \left(1 + \left(\frac{\overline{E_P}}{\overline{P}}\right)^{\omega}\right)^{\frac{1}{\omega}} \tag{8}$$





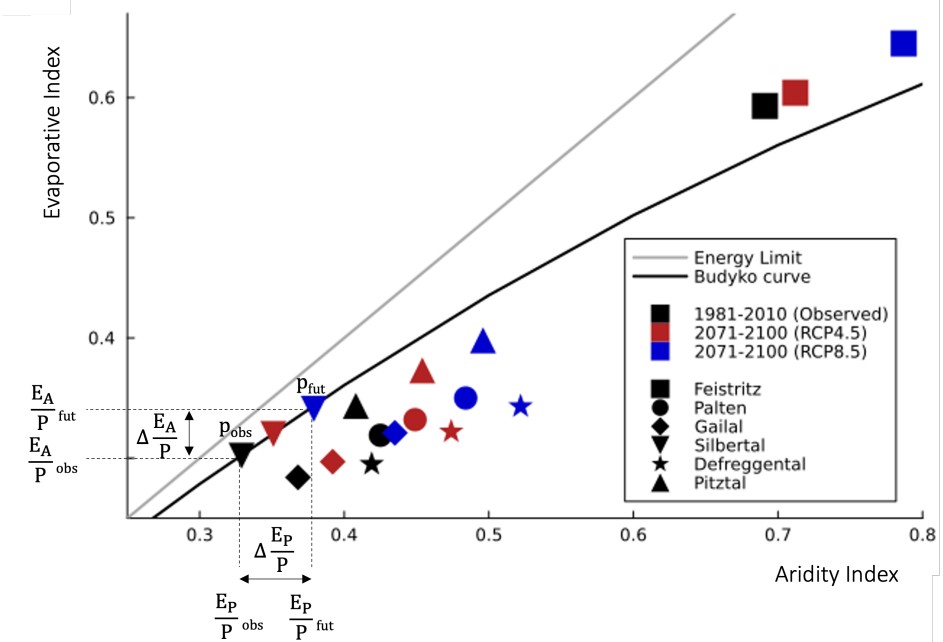

**Fig. 3.** Representation of the Budyko space, showing the Evaporative Index $\frac{E_A}{P}$ and the Aridity Index $\frac{E_P}{P}$ and the energy and water limit. Using observed climate data, $\frac{E_A}{P}_{obs}$ and $\frac{E_P}{P}_{obs}$, a catchment plots on position $p_{obs}$ on the parametric Budyko curve with parameter $\omega_{obs}$. Future climate change, and hence altered input data, result in an altered Aridity Index ($\frac{E_P}{P}_{fut}$), causing the catchment to move along the Budyko curve ($\Delta\frac{E_P}{P}$) towards position $p_{fut}$ that corresponds to a future Evaporative Index ($\frac{E_A}{P}_{fut}$). Mean locations of study catchments and their change over time are depicted in the Budyko space (average over 14 climate models is shown).

### 3.1.3 Estimating future root zone storage capacity $S_{r,clim,fut}$

The combined use of predicted water balance data and the Budyko hypothesis allows us to estimate future root zone storage capacities $S_{r,clim,fut}$. Drawing upon estimates of evaporative ratios from the Budyko framework, past and future long-term mean evaporation can be approximated. In total we derive 29 evaporative ratios per catchment, resulting from the use of observed meteorological data (1981-2010) and future climate projections (2071-2100) of 14 RCMs and 2 different RCPs. By implementing the long-term evaporative indices in the water balance equation, one $S_{r,clim,past}$ and 28 estimates of $S_{r,clim,fut}$ are derived for each vegetation type. Note that utilizing a range of 300 $I_{max}$ values (Section 3.1.1) results in parameter ranges rather than single parameter values for $S_{r,clim}$.

To account for potential biases in the projected climate data, the respective modelled future ($S_{r,clim,fut}$) values are scaled to the difference between observed and modelled past root zone storage parameters, in line with **?** (Supplement S3).





## 3.2 Hydrological model

The influence of a climate-based, time-variant root zone storage capacity parameter on modelled streamflow is analyzed through a process-based, semi-distributed hydrological model (Prenner et al., 2018), as developed by Hanus et al. (2021),
based on the approach proposed by Savenije (2010). This hydrological model represents the dominant rainfall-runoff processes in catchments based on topography and land cover classes. Thereby, the model accounts for the importance of landscape on runoff behavior, whilst retaining a simple model approach. More specifically, four parallel Hydrological Response Units (HRUs) are represented by the model: bare rock, forested hillslope, grassland hillslope and riparian zone. Precipitation input is distributed across different HRUs and trickles down through various subsurface components, visualized by a
bucket system. Water partitioning in the subsurface is governed by specific equations, each with several catchment-specific parameters that require calibration. All parameters remain constant across HRUs, except for the vegetation-dependent parameters: interception storage capacity (Imax) and root zone storage capacity ($S_{r,max}$). These storage capacity parameters vary among individual HRUs to account for differences in vegetation cover. The model schematic and relevant equations are provided in Supplement S1, with a more detailed description available in Hanus et al. (2021).

### 3.2.1 Calibration & Evaluation

In total, the model includes 20 parameters (including parameter $S_{r,max}$) that are initially calibrated on observed streamflow data. For the Pitztal, an additional loss parameter is included, that accounts for artificial water divergence through a pipe system from the catchment. Following the work of Hanus et al. (2021), (Appendix B), all parameters are constrained a-priori based on literature (Prenner et al., 2019; Gao et al., 2014). Additional constraints are provided to ensure that parameter
combinations match the perceptions of the system, e.g. the interception capacity of forests must be greater than that of grass (Gharari et al., 2014).

The model is calibrated using eight objective functions to ensure adequate process representation and to limit model uncertainties (Table 3) (Efstratiadis and Koutsoyiannis, 2010; Hrachowitz et al., 2014). These objective functions include metrics to describe the magnitude and timing of high and low flows (Q, FDC), the memory of the catchment (AC1, AC90),
the partitioning between evaporation and streamflow (RC), as well as the timing of snow cover (SC). All objective functions are equally weighted, as the calibrated model aims to represent overall system dynamics. The overall model performance is then assessed by combining all individual objective functions using the Euclidian Distance ($E_D$) from the perfect model fit, whereas a value of 1 indicates a perfect model (Equation 9; Hulsman et al., 2021).

$$E_D = \sqrt{\frac{\sum_{n=1}^{N} (1 - E_N)^2}{N}} \tag{9}$$

For each catchment, a Monte Carlo Sampling strategy with 3 million realizations is performed, resulting in the same number of possible parameter combinations. These are subsequently referred to as 'calibration parameter sets' (including parameter $S_{r,cal}$). Based on a daily model time step, the calibration was performed over a 20 year period (Oct 1985 - Oct



**Table 3.** Objective functions used to calculate Euclidian Distance ($E_D$) for calibration

| Signature | Abbreviation | Objective Function | Reference |
|---|---|---|---|
| Timeseries of flow | Q | $E_{NSE,Q}$ | Nash and Sutcliffe (1970) |
| | | $E_{NSE,log(Q)}$ | Nash and Sutcliffe (1970) |
| | | $E_{VE,Q}$ | Criss and Winston (2008) |
| Flow Duration Curve | FDC | $E_{NSE,FDC}$ | Euser et al. (2013) |
| Autocorrelation | AC1 | $E_{RE,AC1}$ | Euser et al. (2013) |
| | AC90 | $E_{NSE,AC90}$ | Hrachowitz et al. (2014) |
| Monthly Runoff Coefficient | RC | $E_{NSE,RC}$ | Hrachowitz et al. (2014) |
| Snow Cover | SC | $E_{NSE,SC}$ | Finger et al. (2015) |

2005), with a prior 3-year model warm-up period. Only the best performing 0.01% parameter sets, corresponding to a Euclidean Distance $DE \leq 0.2$, are retained for further analysis to ensure feasible computation time. Through this approach

ill-performing parameter combinations are excluded, while still allowing the model a certain flexibility to account for parameter uncertainties.

The selected 300 parameter sets are subsequently evaluated over the post-calibration period (November 2005 to 2013 or 2015, depending on the catchment) using the objective functions outlined previously (Table 3).

**3.2.2 Testing climate-based root zone storage $S_{r,clim,past}$ for modelling past streamflow**

To test the plausibility of root zone storage capacity estimates inferred from water balance data and to evaluate their influence on the model performance, we replaced the calibrated values $S_{r,cal}$ with the $S_{r,clim,past}$ estimates. More specifically, the calibration parameters $S_{r,cal,forest}$, $S_{r,cal,grass}$ and $S_{r,cal,rip}$ are replaced with a climate-based formulation, respectively $S_{r,clim,forest}$, $S_{r,clim,grass}$ and $S_{r,clim,rip}$. Without any further re-calibration, the model is then re-run for the past period.

This enables a comparison of the model's performance regarding the use of $S_{r,cal}$ and $S_{r,clim}$, based on the 8 hydrological signatures described above.

It is important to note that, due to the introduction of random $I_{max}$ values in the water balance equation, a range is established for each $S_{r,clim}$. Hence, we randomly sample 10 values from each $S_{r,clim}$ range. Hence, the ensemble of 300 calibrated parameters for each vegetation class resulted in 3000 climate-based equivalents, for past conditions and for the

two emission scenarios in the future. Hereafter, models using calibration and climate-based parameter sets are referred to with subscripts $cal$ and $clim$ respectively.





### 3.2.3   The effect of time-variant, future root zone capacity $S_{r,clim}$ on streamflow

To investigate the influence of future adaptation of root zone storage capacity on streamflow, we compare simulations

using stationary $S_{r,clim,past}$ and climate-adapted $S_{r,clim,fut}$. For a comprehensive analysis, we first analyze the change between past and future streamflow, resulting from a changing climate while keeping system parameters stationary, including the climate-based $S_{r,clim,past}$. This run will hereafter be referred to as the stationary model run, using $S_{r,clim,stat}$. Next, we quantify the additional effect of vegetation adaptation by modeling future streamflow using the climate-based, adapted $S_{r,clim,fut}$. This run is hereafter referred to as the adapted model, using $S_{r,clim,adapt}$. Both $S_{r,clim}$ parameter sets

are derived from the calibrated parameter sets, which are calibrated on past stream flow conditions. Without any further re-calibration, the $S_{r,cal}$ parameter is exchanged for the $S_{r,clim,past}$ and $S_{r,clim,fut}$ parameters, respectively.

Next, hydrological change is assessed by examining various streamflow signatures over a 30-year period. These signatures correspond with those explored by Hanus et al. (2021), who delved into the effects of climate change between the past and future on the identical study catchments. Nonetheless, our investigation predominantly focuses on evaluating

the influence of vegetation adaptation on these streamflow signatures, whilst only briefly discussing changes between past and future.

The streamflow characteristics include changes in mean annual discharge, indicating future water availability, mean monthly discharge and both annual and seasonal runoff coefficients. Furthermore, changes in extreme hydrological events are analyzed according to Blöschl et al. (2017, 2019). Specifically, changes in the magnitude of high flows are assessed using

timeseries of Annual Maximum Flows (AMF) and in the context of different return periods. Changes in timing are evaluated using the method of circular statistics (Young et al., 2000; Blöschl et al., 2017), which provides meaningful information about the timing of extreme events despite the turns of the year. However, this method cannot detect a bi-modal flood season, as the average date of occurrence is located in the middle of the flood season. To address this possible non-detection, the relative frequency of AMF occurring within 15 days is also studied. A 15-day time frame allows for the co-occurrence of

different events while providing insight into relatively small changes in AMF over time.

Changes in low flows are assessed using a similar approach, based on the annual minimum runoff over seven consecutive days. Since low flows mainly occur in winter, a moving average from June to May is used to avoid complications with the turn of the year.

## 4   Results & Discussion

### 4.1   Projected changes in climate

#### 4.1.1   Hydroclimatic Change

Annual median temperature and precipitation, averaged over 30 years, are projected to increase for the 2071-2100 period (Figure 4). In spite of some variation between the 14 climate models (GCM-RCM combinations), similar temperature changes were found across all catchments, with multi-climate model median increases around 2-3 °C for RCP 4.5 and 4-5




°C for RCP 8.5 (vs. 1981-2010). The highest median increases are found in the Defreggental and Pitztal. Similarly, broadly consistent increases in precipitation are found across all catchments in the future. Increases are most pronounced for RCP 8.5, where multi-climate model median changes range between +4% in the Gailtal and +9% in the Defreggental. However, the change direction depends on the climate model used, as projections range between -10% and 20% for RCP4.5, with spreads further increasing for RCP8.5.

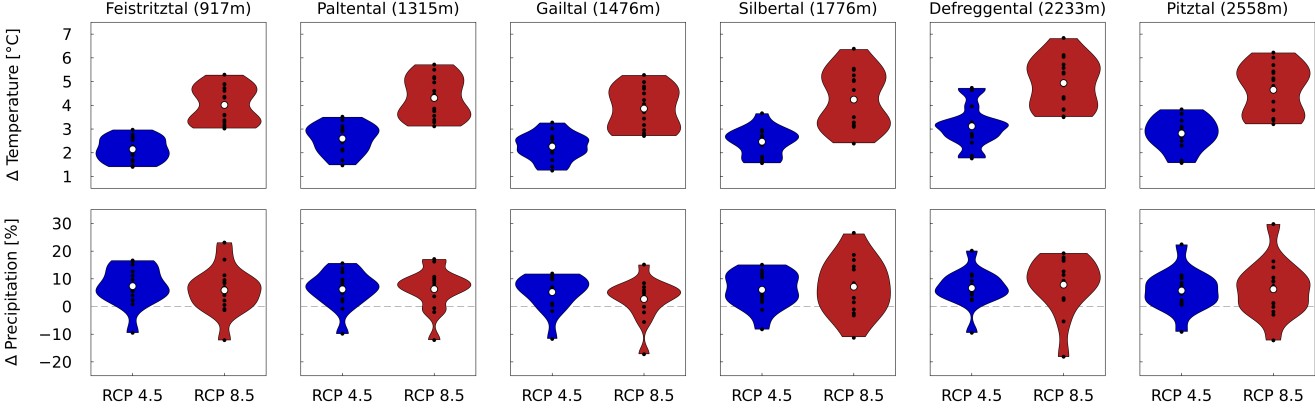

**Fig. 4.** Violin plots, displaying absolute changes in mean annual temperature and relative changes in mean annual precipitation for 2 RCPS and 14 climate models (black dots representing individual RCMs).

### 4.1.2 Future changes in the long-term water balance

For the 1981-2010 period, the Feistritztal and Silbertal show the highest and lowest aridity indices of respectively 0.69 and 0.23 (Figure 3, Supplement S4). This reflects the marked west–to-east gradient in hydroclimatic characteristics of the study catchments. Correspondingly, the evaporative index ranges between 0.28 in the Gailtal and 0.59 in the Feistritztal. Catchment-specific values for $\omega$ range between 1.7 in the Defreggental and 3.0 in the Feistritztal, where higher values for $\omega$

(for a given aridity index) indicate more water use for evaporation. As such, differences in $\omega$ broadly reflect differences in land cover and, in particular, increases in forest cover along that gradient (Table 1, Table S2).

For the 2071-2100 period, aridity indices are projected to increase by ~0.03 in all catchments under RCP 4.5 and up to twice as much under RCP 8.5 (Figure 3, Supplement S4). This indicates drier future conditions with more energy available for evaporation. Again, lowest and highest future aridity indices are found in the Silbertal (0.351-0.379) and Feistritztal

(0.712-0.787) respectively.

Correspondingly, future evaporative indices are expected to increase by ~0.02 in all catchments for RCP 4.5 and will range between 0.60 (Feistritztal) and 0.30 (Gailtal), with further increases for RCP 8.5.





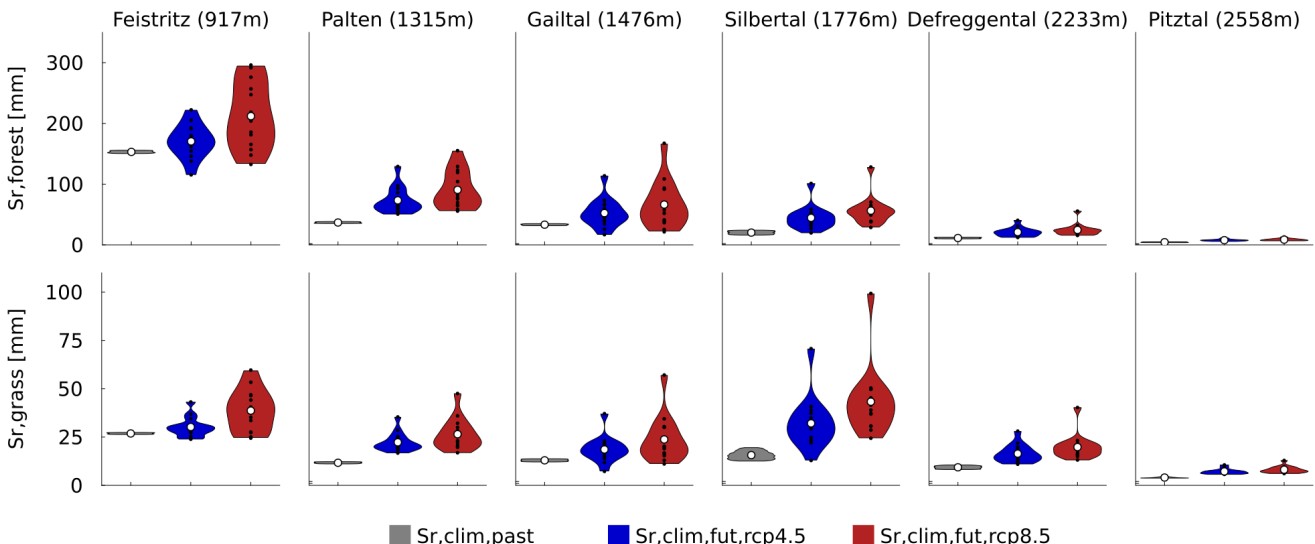

**Fig. 5.** Estimates for $S_{r,clim,past}$ and $S_{r,clim,fut}$ for forest (upper row) and grassland (lower row) landscape elements, inferred from observed past and corrected projected future climate data using the Memory method. A similar figure including calibrated parameter values ($S_{r,cal}$ is included in the Supplementary Material (Figure S5)

## 4.2 Water balance based of root zone storage capacity in the past and future

Using observed past data, the root zone storage capacity parameters range between $S_{r,clim,grass}$ = 5-27mm and $S_{r,clim,forest}$ = 5-155mm (Figure 5). Regardless of the vegetation type, the lowest and highest $S_{r,clim}$ values are found in Pitztal and Feistritztal, respectively. The low root zone values in Pitztal suggest shallow hydrologically active soil depths, which is realistic given that bare rock covers 70% of the catchment area. Conversely, the highest values are found in Feistritztal, where 72% of the area is forested. This relationship between vegetation cover and root zone storage capacity is consistent with

findings by Merz and Blöschl (2004) in a model calibration experiment for 308 catchments in Austria.

The spread in $S_{r,clim,past}$ is below 5mm in all catchments for all vegetation types and directly results from the 300 different interception capacities ($I_{max}$) applied in the Memory method.

For the 2071-2100 period, moisture deficits in the root zone are projected to increase as a result of increased dryness indices and the associated higher evapotranspiration rates (Equation 4 & 3, Figure 3). While $S_{r,clim,forest}$ remains largely

stable in the Pitztal, it increases in the remaining catchments by at least 10mm (100%, Defreggental) and by up to 30mm (75%, Feistritztal) under RCP 4.5, with further increases for RCP 8.5.

Similarly $S_{r,clim,grass}$ increases by 4-20mm for RCP4.5 (6-32mm, RCP 8.5) in all catchments in the future. This translates into relative changes in $S_{r,clim,grass}$ of 14-125%, with the lowest increases in the Feistritztal and Pitztal. The largest increase

in $S_{r,clim,grass}$ is found in the Silbertal.





The spread in estimated $S_{r,clim}$ values increases for the future period for all vegetation types (most outspoken for RCP 8.5), thereby reflecting the uncertainty in the 14 climate models. The spread in $S_{r,clim,past}$ ranges between 5-10mm for forest and 2-8mm for grass, with the largest spread found in the Silbertal. For $S_{r,clim,forest}$ and $S_{r,clim,grass}$, the spread is even more pronounced, ranging between 30-110mm (45-160mm, RCP 8.5) and 18-60mm (28-36mm, RCP 8.5), respectively. 

Notwithstanding these uncertainties, the spread in $S_{r,clim}$ is much smaller compared to the spread in $S_{r,cal}$ arising from calibration (Supplement S5).

### 4.3 Modelled Hydrological Response

Root zone storage capacity parameters, respectively obtained through calibration ($S_{r,cal}$) and the Memory method ($S_{r,clim,past}$), are subsequently implemented in the hydrological model.

**4.3.1 Calibration & Evaluation**

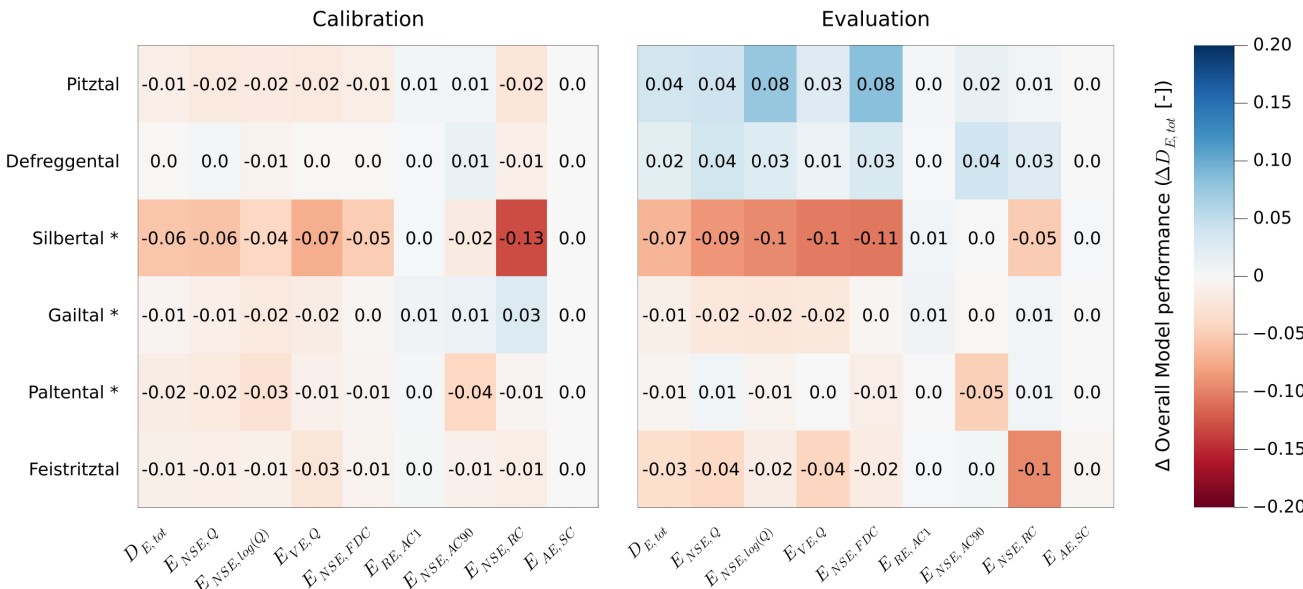

**Fig. 6.** Difference in model performance between the calibrated and climate-based models during the calibration (left) and evaluation (right) period for the overall modelfit ($\Delta D_{E,tot} = D_{E,clim,tot} - D_{E,cal,tot}$) and eight objective functions. Negative values indicate a better model performance of the calibrated model and vice versa. Catchments marked with an aterisk (*) use an 8-year evaluation period instead of 10 years (Section 3.2.1). Table 3 provides a description of the objective functions.

Both models, using the calibrated ($S_{r,cal}$) and water balance-based ($S_{r,clim,past}$) estimates of the root zone storage capacity parameter, broadly reproduce the main features of the observed hydrological response in all study catchments. Using the calibrated parameters, overall model performance ranges between $D_{E,cal,tot}$ = 0.81-0.88 and $D_{E,clim,tot}$ = 0.80-0.87 during the calibration period and remains stable for the evaluation period, with $D_{E,cal,tot}$ = 0.78-0.89 and $D_{E,clim,tot}$ =





0.80-0.85 (Figure S6). Hence, differences in the performance of the two model implementations are limited (Figure 6) with a maximum $\Delta D_{E,tot}$ = -0.06 for calibration and -0.07 for evaluation respectively.

Accordingly, the modelled hydrographs indicate that the short-term flow dynamics (Figure 6, Figure S7-S11) and mean regime curves (Figure 7) are in general adequately captured by the models, regardless of the used parameter set. In some cases, modeled short-term peak flows remain underestimated (e.g. in the Paltental and Defreggental). This underestima-
tion is likely associated with uncertainties in precipitation observations in very localized high-intensity convective rainfall events (Hrachowitz and Weiler, 2011).

Hence we conclude that both the calibrated and climate-based models adequately reproduce the general magnitudes and seasonality in all catchments. The overall good model performance implies that the retained climate-based parameter sets can be used for modelling future streamflow.

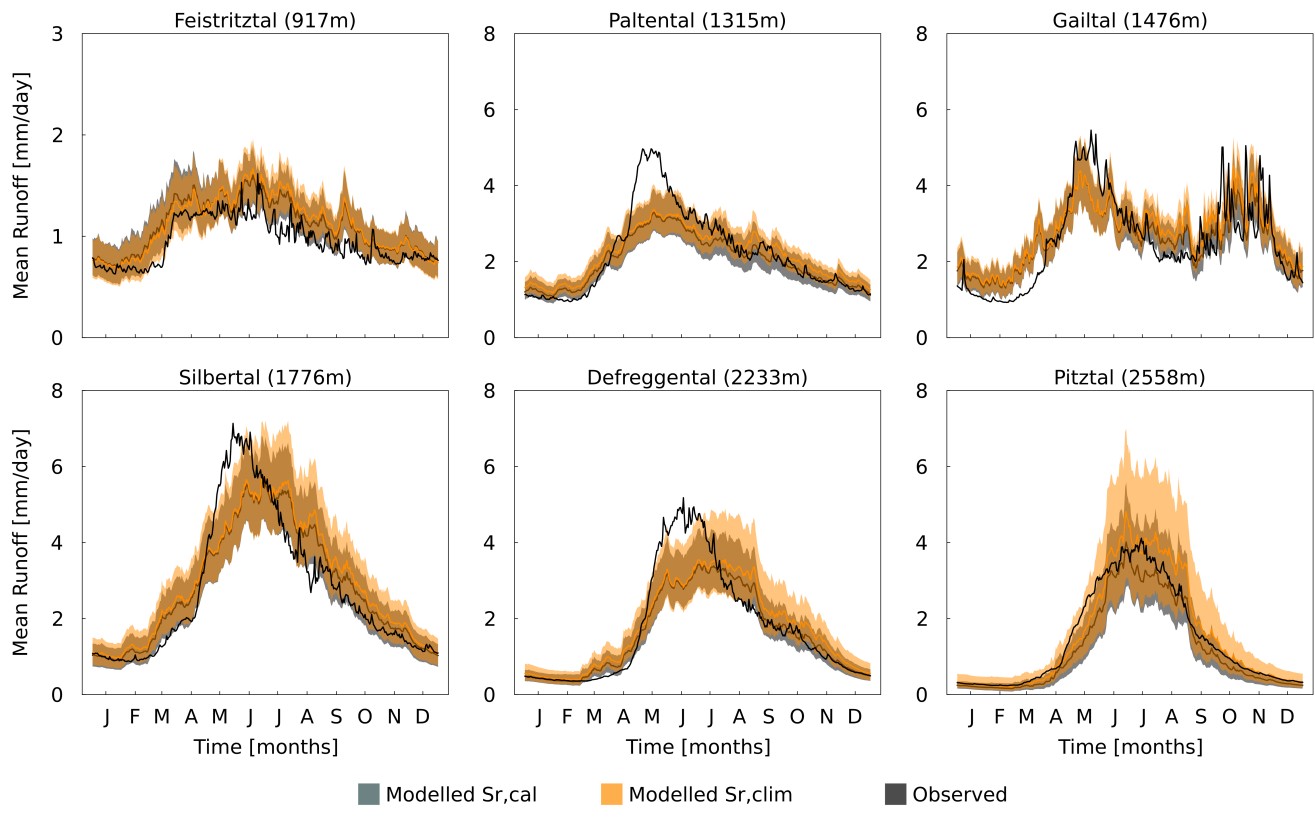

**Fig. 7.** Annual mean regime curves for the six study catchments over the 1981-2010 period. Solid lines represent mean runoff and shaded bands indicate ±1 std.





### 4.4 Future streamflow projections

Future streamflow for the 2071-2100 period is estimated by forcing the hydrological model with projected climate data. To test the impact of a dynamical evolution of the root zone storage capacity on the modelled hydrological response, the modelled streamflow from a model run with stationary $S_{r,clim,stat}$ parameters, obtained from past predicted water balance data, is compared to a model run with an adapted formulation of the $S_{r,clim,adapt}$ parameters (Figure 8), based on projected future water balance data. The two model runs are hereafter referred to as the stationary and adapted model.

#### 4.4.1 Annual discharges

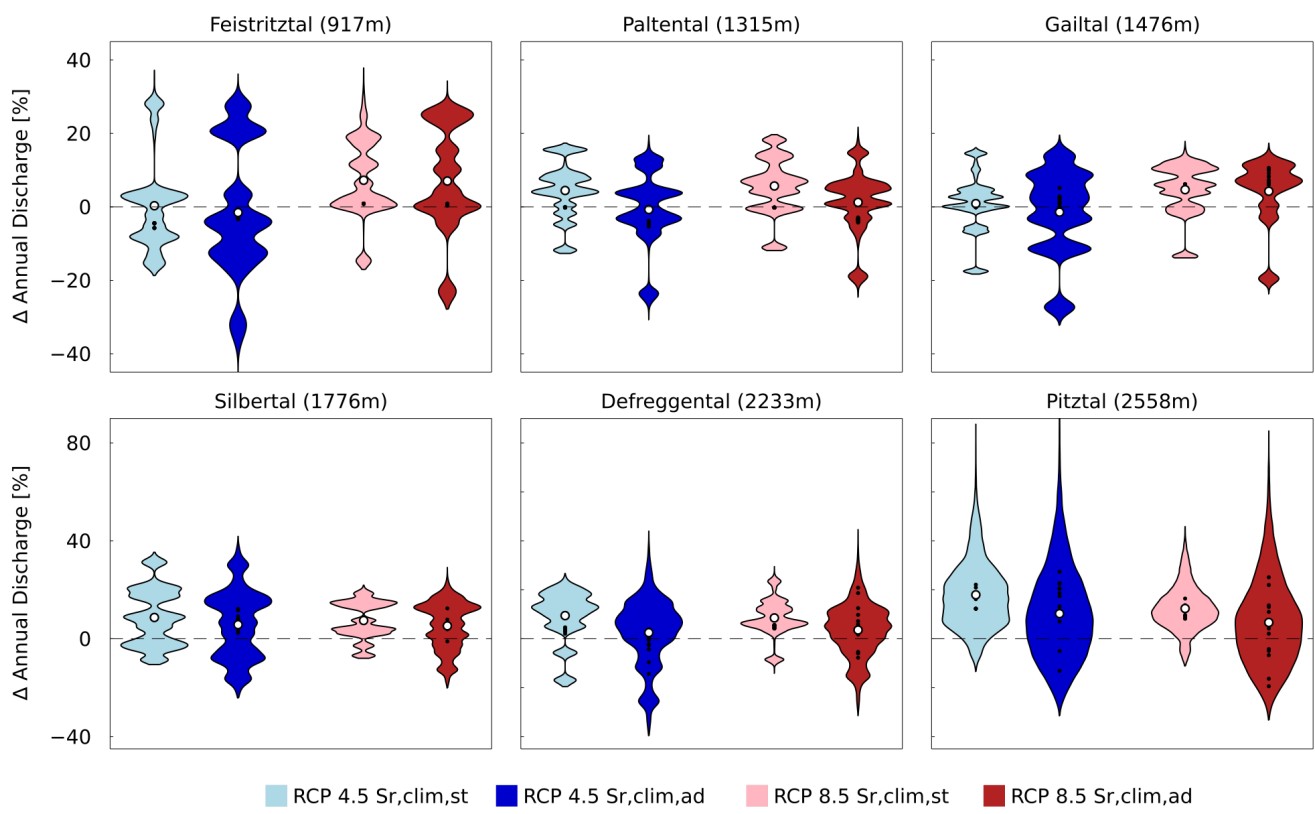

**Fig. 8.** Relative changes in mean annual streamflow for all catchments, using models featuring $S_{r,clim,stat}$ and $S_{r,clim,adapt}$, for 2 RCPS and 14 climate models (black dots representing individual RCMs)

In general, mean annual streamflow in the study catchment exhibits only modest sensitivity to changing climatic conditions (Figure 8, Figure S12). The magnitude and direction of change in streamflow differs per catchment and climate projection used and results from the combined influence of the projected increased annual precipitation and increased





evaporation (Figure 3, Supplement S4). The multi-climate model median temporal change in streamflow across all climate projections varies from -4% (Feistritztal) to +10% (Pitztal) for RCP 4.5, while differences are slightly more pronounced RCP 8.5. Differences in the median modelled temporal change in streamflow between the stationary and the adapted model are very minor, which is in line with findings of Bouaziz et al. (2022).

The modelled mean annual streamflow estimates are also characterized by a relatively high spread (> 100%) for both model runs and both climate scenarios, arising from the uncertainty in the projected hydro-climatic variables from the 14 climate models. Overall and due to the additional uncertainty introduced by the estimation of $S_{r,clim,fut}$, the spread in modelled changes in annual streamflow is somewhat more pronounced for the adapted model, with a spread up to ~120% for the Pitztal.

**4.4.2 Monthly Discharges**

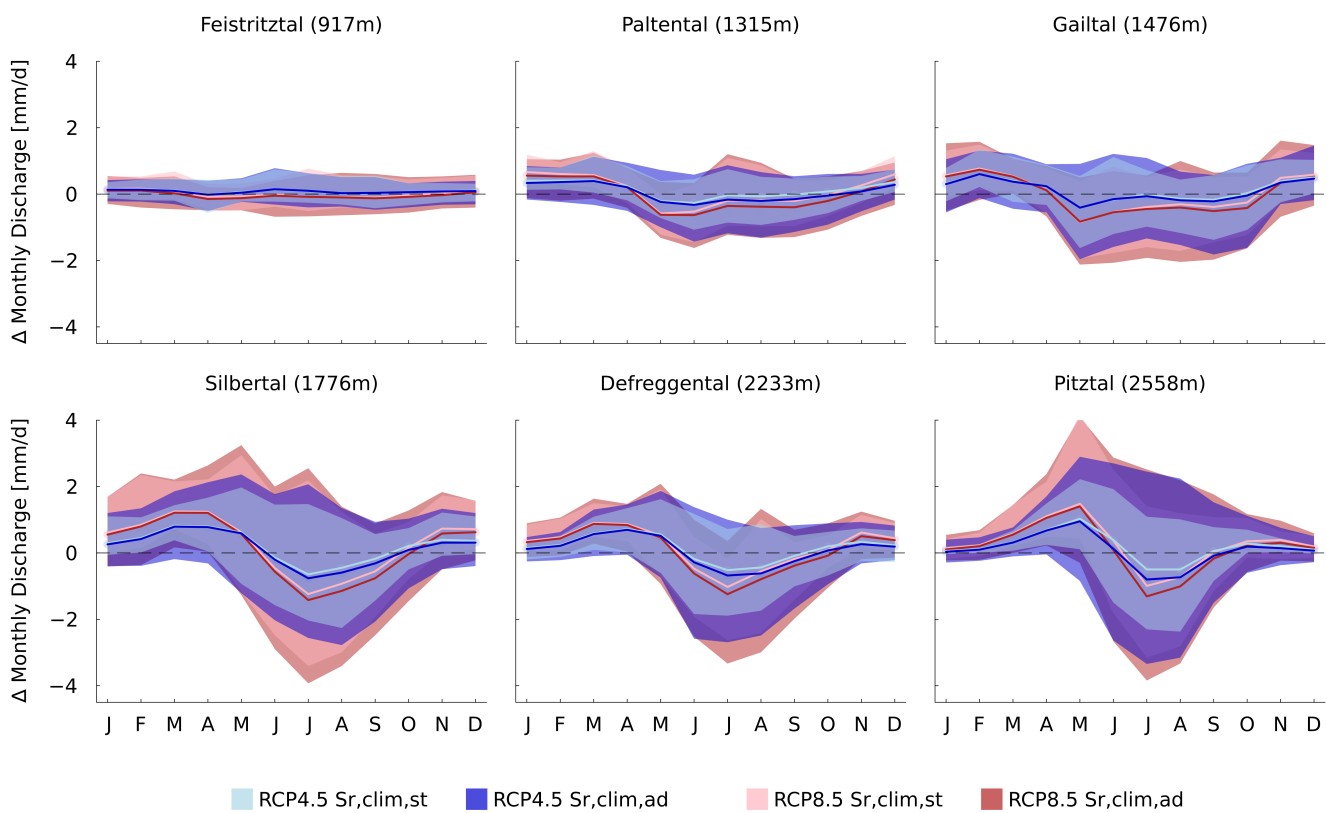

**Fig. 9.** Absolute changes in mean monthly streamflow, using data between the 1981-2010 period and the 2071-2100 period for 2 RCPS and 14 climate models. Solid lines represent mean runoff and shaded bands indicate ±1 std.

The stationary and adapted runs suggest rather consistent seasonal changes in streamflow for all study catchments, albeit with varying magnitudes (Figure 9 & S13). Overall, winter and early spring streamflow is expected to increase by





up to $\Delta Q \sim 1 \text{mmd}^{-1}$ or ~90% at the highest elevations for RCP4.5 and by $\Delta Q \sim 1.5 \text{mmd}^{-1}$ for RCP8.5. In contrast, the projected summer and early autumn streamflow exhibits pronounced reductions by up to ~1 mmd$^{-1}$ or ~20% for RCP4.5

and ~1.5 mmd$^{-1}$ for RCP8.5. The Feistritztal is an exception to this, with little evidence for change in monthly streamflow. In general, the stationary and the adapted model runs give largely equivalent results. The only notable difference is found for mid-summer flow in high elevation catchments, which is projected by the adapted model to be up to ~ 0.5mmd$^{-1}$ (Pitztal) lower compared to the stationary model.

Similarly, no significant difference in the timing of change is found between the stationary and adapted model results.

For both models, the highest future flow increase occurs between February (Gailtal) and May (Pitztal) and the most pronounced reductions occurring between May (Gailtal) and July for the highest elevations. In contrast, for the lower elevation catchments, i.e. the Feistritztal and Paltental, the timing of the maximum change could not be clearly associated to specific months for both model runs. These modelled patterns with a spring increase in flow and summer decrease at high elevations and originate from temperature increases that lead to a shorter snow season (snow-melt period shifts toward earlier

moments in the year).

The spread in results (originating from the climate model induced spread in $S_{r,fut}$) is found to be slightly larger for the adapted model. In the stationary model, the absolute model spread reaches up to ~2 mmd$^{-1}$ (Pitztal) for RCP4.5 and more for RCP8.5, while for the adapted model the spread is around ~3 mmd$^{-1}$, with further increases under RCP8.5).

### 4.4.3 Runoff Coefficient

Both the stationary and adapted models suggest that the climate-model median runoff coefficients ($C_R$) will largely remain stable at low elevations for both RCP4.5 and 8.5. However, higher elevations will experience increased future runoff coefficients with up to $\Delta C_R \sim 0.05$ for RCP4.5 and ~ 0.20 for RCP8.5. Here, the adapted model projects increases that are, with up to ~ 0.05, slightly but systematically lower than those from the stationary model (Figure 10). This supports the findings of Bouaziz et al. (2022) who also reported systematically lower increases in runoff coefficients using a similar adapted model

in the Meuse river basin (France, Belgium). Mechanistically, this pattern can be explained by the fact that future increases in the root zone storage capacity $S_{r,clim,fut}$ in the adapted model lead to more subsurface water being accessible for vegetation. The additional vegetation-accessible water is then used for transpiration and cannot contribute to streamflow anymore. This mechanism is further supported by the observation that for both RCP4.5 and RCP8.5, the largest difference between adapted and stationary model projections of $\Delta C_R$ is found in the Paltental, which is also the catchment with the

largest relative change in $S_{r,clim}$ between past and future. Although the two hydrological model runs provide largely consistent results, the direction of change of the annual runoff coefficient $C_R$ is highly dependent on the climate projection used (Figure 10).

The stationary and the adapted hydrological models also project similar changes in seasonal runoff coefficients. In correspondence with changes in monthly discharge, seasonal runoff coefficients increase in the winter and spring with up to

$\Delta C_R \sim 0.32$ and decrease in the summer months with up to $\Delta C_R \sim 0.15$ as a result of changes in snow-melt contributions (see Figure 10). Slight differences exist between the stationary and adapted model runs in the Paltental and Pitztal, with





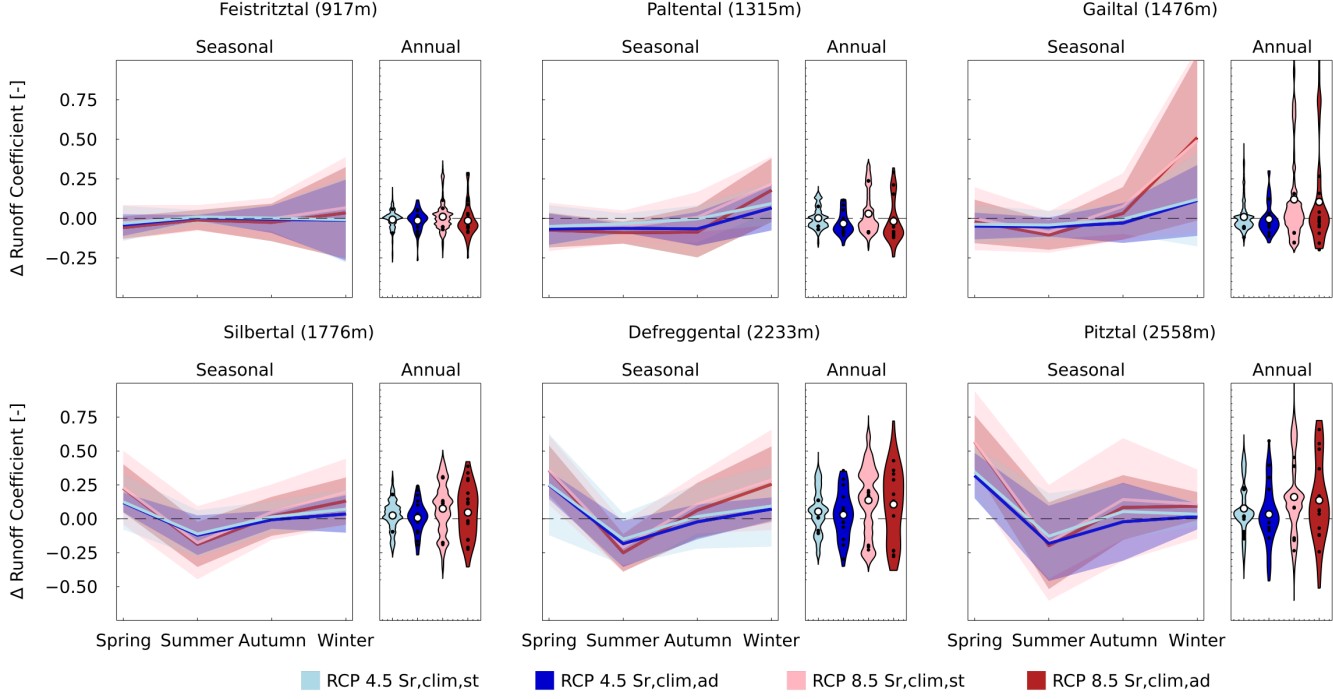

**Fig. 10.** Absolute changes in 30 year average seasonal runoff coefficients, for 2 RCPS and 14 climate models (black dots representing individual RCMs), using the climate-based stationary and adapted model.

the latter projecting autumn $\Delta C_R$ increases that are around 0.1 lower than for the stationary model. For most study catchments, and irrespective of the model run, changes in the seasonal runoff coefficient become more pronounced for RCP 8.5, as temperatures and consequently late-winter and early-spring snow-melt further increase. A similar pattern is found for

relative changes in runoff coefficients (Figure S14).

#### 4.4.4 Annual maxima (timing and magnitude)

According to both the stationary and the adapted model run, the timing of Annual Maximum Flows (AMF) will experience future shifts of up to several weeks (Figure 11, Table S3). While at higher elevations, future AMF is projected to occur ~1 – 3 weeks earlier in the year under RCP4.5, the low-lying Gailtal will experience future AMF to on average occur 3 weeks

later in the year, towards the end of October (Figure 11;Figure 12, right). In both cases the shifts are controlled by changes in the timing of snow accumulation and melt due to increased temperatures. The earlier AMF at high elevations is a result of earlier snow-melt, the later occurrence of AMF in the Gailtal, in contrast, is governed by a higher proportion of autumn precipitation falling as rain instead of snow (Vormoor et al., 2015; Brunner et al., 2020; Hanus et al., 2021). For RCP8.5 the shifts are more pronounced by ~1 – 2 weeks.

Overall, the adapted model projects slightly stronger shifts in timing than the stationary model. For RCP4.5, the shifts



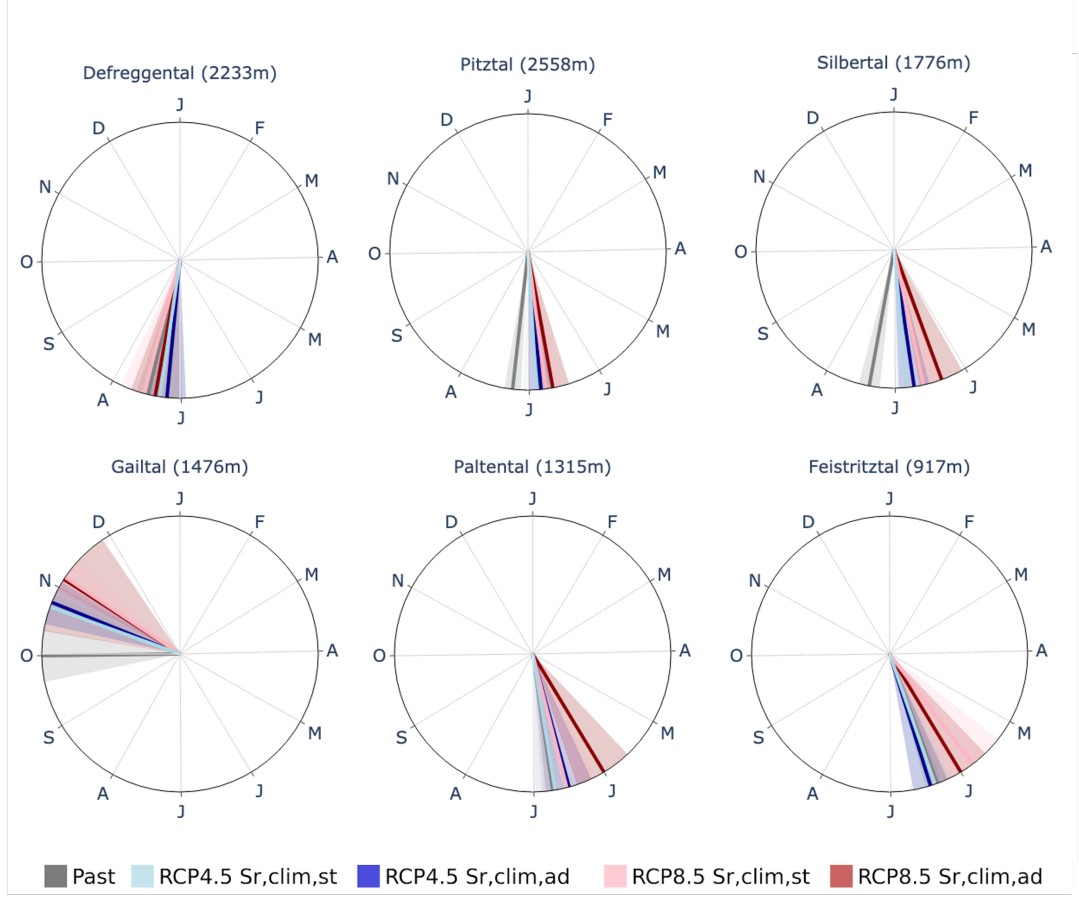

**Fig. 11.** Simulation of mean occurrence of average timing of Annual Maximum Flow over 30 years, for 2 RCPS and 14 climate models, using the climate-based stationary and adapted model. Uncertainty bands of ±1 std are shaded and lines connecting 15-day periods are used to allow for better visualization.

extend over 2-3 days additional days, which increase up to 8 days for RCP8.5. One exception is the Paltental, where the shifts in AMF timing for the adapted model are between 6 (RCP4.5) to 17 days (RCP8.5) more pronounced compared to the stationary model.

A bimodal seasonality of AMF is found for the past for the Gailtal (Figure 12, right) with an AMF in the middle of May and
end of September that is characteristic for the autumn-nival flow regime (Mader et al., 1996; Blöschl et al., 2011). While this pattern is not projected to significantly change in the Gailtal, the stationary and adapted model consistently project that the higher elevation catchments may experience a future transition to a bimodal AMF distribution, in particular for RCP8.5, with a maximum probability of AMF occurrence in the beginning of May and mid-June (Figure 11). This finding suggests, irrespective of the model run, a future extension of the length of the flood seasons in essentially all study catchments,
ranging from 1 month in the Paltental to 2-3 months in the Silbertal and Defreggental.







**Fig. 12.** For each catchment: (Left) Relative change in the 30-year average magnitude of Annual Maximum Flow (AMF). (Right) Mean fraction of occurrences of AMF over 30 years, utilizing a 15-day time window. Uncertainty bands of ±1 standard deviation are shaded, and lines connecting 15-day periods enhance visualization. Both magnitude and timing are assessed for 2 RCPS and 14 climate models (black dots representing individual RCMs), employing both climate-based stationary and adapted models

The stationary and the adapted model models project a similar direction of change for the future AMF and very similar magnitudes of change. Overall, the AMF magnitude is projected to increase by up to ~10% under RCP 4.5 (Figure 12, left), with the lowest increases in the Paltental (~5%), potentially due to the decreased snow-melt in late spring, which offsets the increased precipitation. For RCP 8.5 slightly smaller changes in the AMF magnitude are projected. These more limited





changes are explained by lower snow-melt volumes and higher evaporation rates, associated with higher temperatures, resulting in increased soil storage deficits and hence lower runoff. Compared to the stationary model, the adapted model shows on average 5% lower changes in the average AMF magnitude for both RCP-scenarios. These lower annual mean AMF magnitudes are linked to higher root zone storage capacities and hence lower runoff during extreme events in the future. Only in the Gailtal 15% larger changes in magnitude are projected by the adapted model.

Both stationary and adapted models indicate a consistent increase in absolute AMF magnitude with higher return periods across all catchments (Figure S15)

### 4.4.5 Annual minima (timing and magnitude)

The maximum fraction of occurrences of 7-day consecutive low flows increases with altitude and ranges between 17%
(Feistritztal) and 47% (Gailtal, Figure 13, right). The peak in low flow occurs in between February in lower elevation catchments (Feistritztal) and March for higher elevation catchments (Pitztal).

The stationary and the adapted models project relatively similar future changes in timing, magnitude and spread of low flows. Overall, we project a decrease in the fraction of low flow occurrences in winter months, which is particularly pronounced for RCP8.5, with reductions of ~10% that can reach up to ~20% for the Paltental and Silbertal. This decrease in
fraction of low flows in winter aligns with the findings of Bouaziz et al. (2022), who also report no significant differences in the probability of occurrence of 7-day low flows between stationary and adapted models in the Meuse basin.

Fractions of occurrences of low flows become more spread over the year, with the most decreased seasonality in the Feistritztal. In higher elevation catchments the fraction of occurrences of low flows tends to move towards earlier moments in the year, and thereby the number of months experiencing substantial low flows increases. In the past, the largest
fraction of low flows occurs in February and March. In the future, this moves from January to March, which is related to the earlier onset of the snow-melt season, shifting the start of minimum flows from February to January. In lower elevation catchments an increase in low flow occurrence is projected in the autumn. The highest increase is found in the Paltental, where 13% of the low flow events are experienced in autumn. This relates to decreased summer precipitation and increased evaporation rates. Hence, deficits in the root zone increase, resulting in later discharge release.


  A broadly similar increase of future annual low flow magnitudes is projected by the stationary and adapted models, with a consistent increase in the magnitude of median low flows (Figure 13, left). In high-elevation catchments, this increase is more pronounced for RCP 8.5 and reaches an increase by up to 60% (Defreggental). Although the adapted projections show a high resemblance to the stationary model, change projections are on average 1-5% lower for RCP 4.5 and 4-15% for
RCP8.5. Hence, the adapted model projects a smaller magnitude of future low flows, which is related to increased values of the climate-based adapted root zone storage capacity parameters and related higher water retention in the root zone, resulting, in turn, in reduced runoff volumes.

The direction of change is very uncertain for low flows in the Feistritztal, Paltental and Gailtal. While for the stationary





**Fig. 13.** For each catchment: (Left) Relative change in the 30-year average magnitude of 7-day consecutive lowest flows. Note the different scales for high- and low-elevation catchments. (Right) Mean fraction of occurrences of yearly lowest 7-day consecutive flow in 30 years, using a 15-day time window. Uncertainty bands of ±1 standard deviation are shaded, and lines connecting 15-day periods enhance visualization. Both magnitude and timing are assessed for 2 RCPS and 14 climate models (black dots representing individual RCMs), employing both climate-based stationary and adapted models

model projected spreads in magnitude of low flows vary between 80% (Paltental) and 240% (Pitztal) are found, the adapted results in a larger spread, between 110% and 240% respectively (for RCP 8.5).





## 5  Broader implications and outlook

A broad consistency was found when modeling streamflow with either either a stationary or an adapted root zone storage capacity parameter. While projections show significant changes in the future root zone storage capacity of forests and grasslands, with potential increases of up to 100% and 45% respectively compared to historical levels, accounting for this by adjusting the associated model parameters only results in a limited decrease of projected streamflow and differences in annual mean streamflow projections are close to zero. This also holds for differences in 7-day consecutive low flows and fractions of occurrences of high and low flows, for which differences are very limited for all catchments. Differences in the annual runoff coefficient and annual maximum flows projected by the stationary and adapted model runs are around 5%. However, on a seasonal scale, the model differences can be larger, with the adapted model projecting up to -10% lower monthly discharges and seasonal runoff coefficients. In general, in our study, the differences between the model projections are the smallest for the lower elevation catchments and increase with elevation. This pattern corresponds to the projected magnitude of change between past and future streamflow, which is particularly pronounced in higher elevation catchments..

Our findings point into the same direction of change as previous studies that implemented vegetation dynamics in hydrological models, although our modelled changes are generally smaller. Bouaziz et al. (2022) modelled changes in annual mean (-7%), annual maximum (-5%) and 7-day low (0%) flows when replacing calibrated, stationary root zone storage capacity parameters with adapted estimates. Merz et al. (2011) also showed that models with stationary parameters overestimate median average annual and maximum streamflow by respectively +15% and +35%. Similarly, Duethmann et al. (2020) and Speich et al. (2018) showed model overestimation to reduce when accounting for evolving vegetation dynamics. As a response to changes in vegetation, evaporation rates were found to increase from +4% (Bouaziz et al., 2022) to as high as +67% (Merz et al., 2011), indicating substantial variability on the order of magnitude of future changes to occur in adapted modeling. Although the magnitude of change in our study is less pronounced, a similar direction of change is distinguished: runoff coefficients are found to decrease, inferring higher evaporation rates.

The six studied catchments are all relatively humid (Figure 3) and are therefore characterized by shallow hydrologically active root zones (Merz and Blöschl, 2004). Also, the effects of rising precipitation and higher temperatures typically balance each other, thereby limiting the potential for significant expansion in root zone storage capacity. Hence, to obtain a more general conclusion on the advantages and limitations of this approach, the methodology should be further explored in a broader range of climates. In particular, it is plausible to assume that catchments in more arid environments are likely to experience larger changes in the root zone storage capacity that may further result in a stronger contrast between adapted and stationary model results. This hypothesis is supported by our findings for the Paltental, which is the catchment with the most pronounced temporal change in the root zone storage capacity, resulting in relatively large future differences between the stationary and adapted models. Similarly, Bouaziz et al. (2022) and Merz et al. (2011) found more pronounced differences in streamflow of +34% and +67%, respectively, in response to larger increases in root zone storage capacity under a 2K warming scenario. However, both studies apply averaging over multiple catchments with a large





climatic gradient, which might result in larger changes in root zone storage capacity. More specifically, Merz et al. (2011) showed that averaging over only the relatively humid catchments results in lower increases in root zone storage capacity of +46%, compared to 67% when also including drier catchments.

It is likely that changes in root zone storage capacity (the only ones we consider in this study) co-occur with changes in other system characteristics, influencing the liquid water input and in turn the root zone storage capacity. Merz et al. (2011)

account for this by coupling a simulation of a land-surface model to a hydrological model, which partially explains larger differences in modeled streamflow compared to our study. Future research should investigate the effect of changes in $S_R$ on other model parameter dynamics on future projections and the use of the Budyko framework.

## 5.1    Uncertainty & Limitations

Our study quantifies changes in modeled hydrological response, whilst relying on the combined use of the Memory method and a parameterized Budyko framework. The various assumptions on which our study is based, result in associated uncertainties. First, the Memory method relies on the assumption that vegetation will - and has had the time to - adapt to prevailing climate conditions and does so in compliance with the dynamic equilibrium described by the Budyko framework. Gentine et al. (2012) for instance showed that vegetation eventually adapts to satisfy its water needs, which is reflected in

the scattered pattern of catchments worldwide plotting around the Budyko curve (Troch et al., 2013). Yet, considering the unprecedented scale and rate of current climate change (Gleeson et al., 2020), it is unclear how ecosystems will cope with these changed conditions. In line with this, the assumed return period of dry periods that can be bridged through root zone adaptation in the future is uncertain and can have a considerable effect on the estimated future root zone storage capacity. The severity of influence depends on the magnitude of the used return period, which follows from the logarithmic

shape of the GEV distribution. In addition, future changes in long-term mean runoff are estimated from a parameterized Budyko equation, which assumes that the catchment-specific parameter $\omega$ represents biophysical features of the catchments and hence changes in response to changed aridity. However, recent work of Reaver et al. (2022) and Berghuijs et al. (2020) illustrates the need for careful and considerate use of the parameterized Budyko equation in changing systems.

Secondly, we do not explicitly account for the impact of climate change on catchment functioning and other vegetation

characteristics (Seibert and van Meerveld, 2016), such as adaptation vegetation water use towards water availability (Zhang et al., 2001) and increasing $CO_2$ concentrations resulting in a water-saving response and increased productivity (Keenan et al., 2013; Van der Velde et al., 2014; Ukkola et al., 2016; Jaramillo et al., 2018).

Thirdly, we do not model changes in maximum interception storage (Calder et al., 2003), since Bouaziz et al. (2020) showed impacts to be relatively minor. Changes in both natural and human-induced future land use and land cover have

not been considered here despite their potentially significant influence on the hydrological response (e.g. Jaramillo and Destouni, 2014, Nijzink et al., 2016; Hrachowitz et al., 2021) that would, amongst others, result in vertical movements in the Budyko space (Bouaziz et al., 2018; Jaramillo et al., 2018).





# 6 Conclusions

Understanding the non-stationarity of hydrological systems in a changing climate is a major challenge in hydrology (Blöschl, 585 2010). Despite the importance of non-stationarity, a knowledge gap currently exists concerning the meaningful implementation of system changes in hydrological models. In this context, process-based approaches can help bridge this gap. These approaches apply a holistic consideration of the co-evolution of soil, climate, and vegetation, thereby enhancing the understanding and modeling of non-stationary hydrological systems. Our study introduces an exploratory top-down framework for describing adjustments in root zone storage capacity under the influence of climate change. This framework serves as 590 an initial step towards understanding the non-stationarity in $S_r$ and the potential effect this will have on future streamflow projections for six catchments in the Austrian Alps.

We found that catchment dryness and evapotranspiration increase in the future, an effect that is particularly pronounced under strong warming (RCP 8.5). As a result, future mean root zone storage capacities are found to increase for all catchments by up to 30 mm (+19%, $S_{r,clim,forest}$, Feistritztal, RCP4.5). However, it's worth noting that these increments in $S_{r,clim}$ 595 values also entail a significant range, which can influence the direction of change. Overall, replacing stationary root zone storage capacity parameters with adapted estimates generates broadly consistent model results in the study region: Our results show a consistent pattern of change in future streamflow, though the adjusted model predicts slightly lower streamflow projections, with variances in annual mean and extreme flows averaging around -5%, and up to -10% for runoff coefficients and monthly discharges. The differences were found to be highly dependent on the catchment, time of the year, 600 used climate model and emission scenario, indicating the need to further study non-stationarity for other settings.

Overall, our findings suggest little to no support for the hypothesis that vegetation adaptation, as assessed through the water-balance method, significantly alters the hydrological response within the catchments under study. This implies that adjusting the root zone storage capacity parameter to accommodate vegetation adaptation may not be crucial for accurately projecting future streamflow in our temperate-humid study areas. However, it is important to acknowledge the con- 605 siderable uncertainty ranges inherent in our results. Furthermore, additional research is warranted to illuminate the effects in regions where more pronounced vegetation adaptation, such as larger absolute changes in root zone storage capacity, is anticipated in response to a changing climate.

*Code and data availability.* Hydro-meteorological data were provided by the Hydrological Service Austria and Central Institute of Meteorology and Geodynamics (ZAMG). The climate simulation data were produced by Wegener Center for Climate and Global Change, 610 University of Graz (Douglas Maraun and Matt Switanek). The model code is written in Julia (https://julialang.org/) and available on GitHub (https://github.com/mponds01/HBVmodel).

*Author contributions.* MP: developed the model code, performed simulations, analyzed and visualized data, wrote the original manuscript draft, and processed review comments. SH: Developed the model core and contributed to writing through review and editing. HZ, MtV,





GS: Contributed to writing through review and editing. RK: Provided the observed streamflow data for the study. MH: Supervised the research and contributed to writing through review and editing.

*Competing interests.* MH and MtV are members of the editorial board of HESS.

*Acknowledgements.* MP and HZ acknowledge the funding received from the European Research Council (ERC) under the European Union's Horizon Framework research and innovation programme (grant agreement No 101115565; 'ICE3' project)





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
