# Peer review of "Adaptation of root zone storage capacity to climate change and its effects on future streamflow in Alpine catchments: towards non-stationary model parameters"

_Hydrology and Earth System Sciences, 2024_

## Author Response (AR1)

Reviewer #1

Upfront we would like to express our sincere gratitude to the reviewer for the time and effort invested in reading our manuscript. We highly appreciate the critical, yet very constructive and insightful comments. Below we provide clarifications in detailed replies to all comments.

*Comment:*

*Ponds et al. explore how a non-stationary model parameter (root zone storage capacity) will impact streamflow in humid (energy-limited) catchments under future climate scenarios. My understanding is that previous work from the coauthors has used the same model and the same data to explore the impact of future climate scenarios in the same catchments (Bouaziz et al. and Hanus et al.). Thus, this contribution's impact hinges on what we learn about nature/how the world works by exploring the implications for streamflow of changing a single parameter (S_R) in an existing hydrological model in a particular climate type.*

Reply:

We realize that the description of how our study is different to previous ones has not been sufficiently clear in the original manuscript. Therefore we would like to clarify two points.

Firstly, the reviewer is right in assuming that our previous work in Hanus et al. (2021) has explored effects of a changing climate on streamflow, based using the same model in the same study catchments in the Central Alps. The critical difference in that earlier study was that it did not account for potential changes in the role of vegetation. As such the predictions of Hanus et al. (2021) were based on a stationary model parameter of root zone storage capacity $S_r$, as is currently still common practice in the vast majority of studies that aim to predict the climate change effects on hydrology (see also comment/reply below).

In contrast, Bouaziz et al. (2022) investigated potential effects of climate change on $S_r$ in the Meuse basin, a hydro-climatically substantially different region in NW-Europe. While also an energy-limited basin, it is characterized by seasonal precipitation and energy (i.e. $E_P$) signals whose amplitudes are out-of-phase: a winter rain regime, in which with the highest flows in winter and the lowest in summer. The alpine region of the current study, instead, is a snow-dominated regime, where the amplitude of seasonal water supply, consisting of snow melt, glacier melt and rainfall is much more in-phase with the seasonal energy signal and where high flows occur in the summer and low flows in the winter months.

Secondly, while for essentially all other model parameters it is at this point problematic if not at all impossible to quantify their future changes in a meaningful way for catchment-scale applications, Bouaziz et al. (2022) further outlined a method for future estimates of the root zone storage capacity. Leveraging the potential of this method in the current manuscript, we indeed analyse the "*implications for streamflow of changing a single parameter*", which is actually the only catchment-scale (or "effective") parameter for which we can at the present do so in a systematic and plausible way.

*Comment:*

*I am not as familiar with the large literature on process-based hydrologic models, but while reading I was left with the impression that surely this knob must have been turned in prior studies? Could the authors please make explicit the novelty of exploring the impacts of changing this parameter, and summarize previous works that have done so (or state that it has indeed never been done)?*

Reply:

There are some studies that have analysed time-variable, catchment-scale model parameters *ex-post* by time dynamic calibration over multiple past periods (e.g. Wagner et al., 2003; de Vos et al., 2010; Merz et al., 2011; Stephens et al., 2019) or by linking parameters to past time-series of remote sensing data, as was for example done by Duethmann et al. (2020) who used temporal variability in remotely sensed NDVI signals to correspondingly dynamically adjust the vegetation surface resistance parameter in their model.

However, an *ex-ante* and thus forward extrapolation of catchment-scale parameters remains challenging. This is foremost due to a lack of robust mechanistic or statistical relationships between catchment characteristics, their evolution over time and model parameters as, amongst others, pointed out by Wagener (2007), Fatichi et al. (2016) or Stephens et al. (2021). In other words, even if we knew (which we rarely do) how certain catchment characteristics, for example vegetation composition, will change, we still have insufficient means to quantify how this will affect model parameters. Note, that existing spatial parameter regionalization schemes that seek to identify relationships between catchment characteristics and parameters largely rely on static catchment attributes (e.g. soil types) or past time series of some observed variables (see also above; e.g. LAI/NDVI; Samaniego et al., 2010), making them unsuitable for future estimations.

For that reason, the vast majority of studies that use catchment-scale models to analyse the effects of climate change do so by using either (a) stationary model parameters or (b) different scenarios of how catchment characteristics, for example vegetation composition will change, based on a sensitivity analysis with *ad hoc* assumptions of how these changes may affect model parameters (e.g. Bormann et al., 2007; Huisman et al., 2009; Bulygina et al., 2012; Guimberteau et al., 2017; Pechlivanidis et al., 2017; Gaur et al., 2021; Padulano et al., 2021). This lack of a systematic, time-variable adaptation of catchment-scale parameters applies not only to hydrological models but also to the vast majority of land surface schemes of climate models as recently pointed out by van Oorschot et al. (2021, 2023).

The methodology outlined by Bouaziz et al. (2022), argues that parametric formulations of the Budyko framework (e.g. Tixeront, 1964) constitute the rare case of a robust semi-empirical relationship, strongly supported by mechanistic reasoning (e.g. Porporato et al., 2004; Gentine et al., 2012), that can facilitate the estimation of future changes to a catchment-scale model parameter, i.e. the root zone storage capacity.

The novelty of our study is that it is the first analysis to systematically estimate future adaptations of catchment-scale root zone storage capacity as model parameter based on that methodology and to quantify the cascading effect thereof on future predictions of streamflow in an alpine environment.

We will further clarify this in the revised manuscript.

*Comment:*

*Many previous works have explored the interactions between root zone storage capacity, aridity, and water partitioning (e.g., Porporato et al., 2004, and references cited therein). These studies have emphasized the distinct dynamics that are likely to occur under arid vs. humid regimes. I think that the restriction of this contribution to a narrow range of aridity (humid, energy-limited environments) limits the scope of the findings and usefulness of the study, and suggest that the impact of the paper would be significantly greater if a diversity of climate regimes were explored.*

Reply:

We completely agree that the iconic work of Porporato et al. (2004) and subsequent studies provide insightful descriptions of the differences between different types of environments. While we agree that an exploration of the issue across a wide spectrum of environments can provide a wider perspective, this would be an entirely different analysis, which would necessarily come at the price of much less detail. The Alps being the major source of water for much of central Europe, we deliberately chose to confine our study to this regional scale. This allowed us to analyse climate effects on hydrology in a more comprehensive way and from different aspects, including seasonal water supply as well as timing and magnitudes of extremes (i.e. floods and low flows), all of which play different and sometimes even contrasting roles for developing efficient future water resources management strategies in the region.

We will explain our choice in more detail in the revised manuscript.

*Comment:*

*The paper is very long and complicated for what it is - the exploration of how a parameter changed in a model compares to previously published work with the same model. I would suggest that the authors consider ways to simplify and shorten the work where possible.*

Reply:

We agree that some sections in the manuscript are lengthy and unnecessarily convoluted. We will shorten these parts and make the manuscript more easily readable.

However, we also want to reiterate that this study does not only analyse "*how a parameter changed in a model compares to previously published work*". Instead it is less about how the parameter changed (p.16 in the original manuscript) but rather about the consequences of that change (p.17-26). Using the outline of Bouaziz et al. (2022), the study is, to our knowledge, the first one to systematically infer future changes of a model parameter based on a robust relationship (i.e. the Budyko framework) and to quantify the effects thereof on future streamflow.

We will further clarify that in the revised manuscript.

*Comment:*

*The study adopts the assumption that roots will be able to grow as much or as little as needed to obtain water to overcome droughts of a particular recurrence interval, without consideration to how substrate may limit rooting. The studied include glaciers (and presumably, large expanses of*

*exposed, relatively fresh bedrock, whose area grows under the future climate scenarios in which warming has resulted in glacier retreat). It is not clear that this assumption is realistic for the study catchments.*

Reply:

Two of the study catchments indeed include glaciers, accounting for 1.5 and 18% of the respective catchment areas (Table 1 in the original manuscript). The glaciers are represented in the model HRU referred to as "Bare Rock/Sparsely vegetated" (Supplementary Material Fig.S1). The past and estimated future glacier retreat, as described in Section 2.2.1 in the original manuscript, is accounted for by changing areal fractions over time as described by variable $A_{GI}$ [-] (Supplementary Material Tab.S1). The fraction of the HRU "Bare Rock/Sparsely vegetated" that is covered by glacier ($A_{GI}$) is assumed to have a perennial snow pack that allows continuous melt on days with temperatures exceeding the threshold temperature $T_{Thres}$, thereby assuming a *de facto* "infinite" snow water storage. The fraction not covered by glacier ($1-A_{GI}$) can only generate snow melt as long as a seasonal/transient snow pack is present.

In absence of significant vegetation, this bare rock HRU is characterized by very low root zone storage capacities $S_{R,bare}$ < 10 mm that also include surface/rock interception. Ranges and best parameter values (3000 per catchment) per HRU are displayed respectively in Figure 1 below.

Please note that for the bare storage capacity parameter, no adjustments are made from the calibration, which is why the subscript remains $S_{R,cal,bare}$. In contrast, the other parameters are recalculated using the memory method, represented as $S_{R,clim,....}$

We will clarify all of the above in the revised manuscript.

[Figure]

*Figure 1- Overview of 3000 best parameter ranges by HRU by catchment. Please note: axis vary by HRU.*

*Comment:*

*There is a strong coupling between the omega value in the Budyko framework and the root-zone water storage capacity, which has been explored extensively in the literature (including by the references cited in this contribution). It was unclear to me how the assumption of a static omega under a future climate did not result in a circular or forced outcome when then determining how plants would 'resize' their root zone water storage capacities. I may have missed something fundamental - but by forcing the omega value to be the same, under a warmer (more arid) future, the future water partitioning is being forced as well under the Budyko framework. It is therefore unclear how what was being studied (the impact on streamflow under a future climate with a different storage capacity) was independent (not baked into) the methodology that forced streamflow to behave a certain way (according to Budyko, with a fixed omega value). Alternatively, if the outcome for streamflow is not already predetermined by assuming the fixed omega value, then another issue appears: how is the storage capacity allowed to independently evolve (if, as already established by previous studies, it is strongly coupled to the aridity index and evaporative index). In other words, if omega is a function of storage capacity, and you fix omega, how are you exploring a dynamic storage capacity?*

Reply:

We completely agree with the reviewer on the strong coupling. Assuming validity of the Budyko framework, we know that long-term mean aridity is a first order control on long-term mean $E_A/P$ (and thus on $E_A$ and Q). Forcing a fixed omega does therefore indeed force long-term mean future water partitioning. However, while this holds for long-term averages, there is much weaker coupling at the shorter time scales that are the focus of our study. Little (if anything) about stream flow dynamics and associated the magnitudes and occurrences of floods and low flows can thus be inferred from that partitioning.

We similarly agree that omega is a function of root zone storage capacity. However, it is not *exclusively* a function of root zone storage capacity (and thus of $E_A$) but also of other factors, such as vegetation water use efficiency (e.g. Gentine et al., 2012). Thus following Bouaziz et al. (2022), the explicit assumption here is that fixing omega is equivalent to keeping these other factors constant, while increased $E_A$ in more arid future conditions is sustained by increased root zone storage capacities.

We realize that we have not described and discussed this in sufficient detail in the original manuscript. We will add more detailed information and discuss the implications of the assumption in the revised manuscript.

*Comment:*

*There is a crucial methodologic step that is described in one sentence but without sufficient detail to understand what was actually being done: "By implementing the long-term evaporative indices in the water balance equation, one Sr,clim,past and 28 estimates of Sr,clim,f ut are derived for each vegetation type". Please elaborate.*

Reply:

We agree, that this was not well explained in the original manuscript.

The procedure follows several distinct steps:

Estimate past root zone storage capacity $S_R$

1. (1) Use observed past long-term means of time series of precipitation P and streamflow Q to estimate past long-term mean transpiration $E_R$ (Eqs. 1-2). Note, that the total evaporative fluxes are combination of transpiration and interception evaporation, $E_A=E_R+E_I$
2. (2) Estimate time-series of past daily $E_R(t)$ by temporally redistributing the mean $E_R$ according to the time series of daily $E_P$ (or vice versa: rescale $E_P(t)$ so that the mean of the rescaled time series equals to mean $E_R$; Eq.3)
3. (3) Compute daily time series of storage deficits as $S_{R,D}(t) = \sum(P_E(t) - E_R(t))$ and determine the highest deficit for each year as $S_{R,D,yr} = max(|S_{R,D}(t)|)$ (Eqs.4-5)
4. (4) Fit a GEV distribution to the set of highest annual deficits $S_{R,D,yr}$ and estimate the root zone storage capacity $S_{R,clim,past}$ as maximum annual deficit over a specific time interval (or return period; here 20 yrs for forest, 2 yrs for grassland)

To estimate future root zone storage capacity $S_R$, estimates of future transpiration $E_R$ are needed. As no observations of future streamflow are available, the following procedure is adopted:

(5) Determine the past long-term mean catchment position defined by $E_P/P$ and $E_A/P$ in the Budyko framework and fit the associated parameter ω (Eq.6).

(6) With projected long-term mean future P, $E_P$ and thus $E_P/P$ and assuming a constant ω, long-term mean future $E_A/P$ and thus $E_A$ are estimated for each of the 28 future climate scenarios.

(7) With his long-term mean future estimate of P and $E_A$ steps (1)-(4) are repeated to estimate $S_{R,clim,fut}$ for each of the 28 climate scenarios.

As pointed out by the reviewer, the assumption of a constant ω may not be fully warranted, due to the mutual interactions of aridity, seasonality and root zone storage capacity.

However, and to put this into perspective, the vast majority of studies up to now use the past $S_{r,clim,past}$ for any future climate impact analysis. This entails the strong assumption of a jump in ω, which may be associated to various factors such as changes in water use efficiency or vegetation density. For example, under more arid conditions (i.e. higher $E_P/P$), vegetation then transpires the same fraction of precipitation (i.e. $E_A/P$) than under previous more humid conditions. In that example, there is no fractional increase in transpiration, because vegetation, for example, does not need more water under more arid conditions ("water use efficiency") or there is less vegetation because individual plants died as they could not satisfy increased transpiration requirements due to insufficiently large root systems ("root zone storage capacity"). Yet, this is largely not what we are seeing in the Budyko framework, where long-term mean $E_A/P$ generally increases with increases in long-term means of $E_P/P$. Although over time catchments do not strictly follow their specific curves defined by ω (e.g. Reaver et al., 2022), several recent studies have shown that the deviations from their curves (and thus changes in ω) over time remain very limited (e.g. Ibrahim et al., 2024; Tempel et al., 2024; Wang et al., 2024)

We will clarify this in the revised manuscript.

*Comment:*

*(6) The model employed has so many free parameters and processes and things involved (different land cover classes behaving differently, e.g.), that the suspicion arises of whether we can expect to actually isolate the desired impact on streamflow of the term of interest (S_R). Can the authors convince the reader that the other numerous features of the model are not drowning out a signal?*

Reply:

Indeed, elevated degrees of freedom, related to high numbers of parameters, do pose a challenge to distil meaningful signals in models. In a deliberate decision to limit this problem, the model was, adopting a multi-objective strategy, simultaneously calibrated to 8 distinct objective functions, thereby forcing the model to simultaneously reproduce 8 complementary signatures of the hydrological response. This approach is very effective in reducing false positives (i.e. parameter sets falsely accepted as feasible; e.g. Gupta et al., 2008; Efstratiadis and Koutsoyiannis, 2010; Hrachowitz et al., 2014) and the risk of "getting the right answers for the wrong reasons" (Kirchner, 2006).

Overall, the model showed good skill to simultaneously reproduce all 8 signatures in all study catchments as shown in Figure S6 in the Supplementary Material.

We will clarify that in the revised manuscript.

*Comment:*

*Line 59 - missing reference. also on line 251*

Reply:

This will be corrected.

*Comment:*

*Line 80-81. It is important to emphasize that deficit-based approaches can only constrain (i.e., provide a lower bound or minimum estimate) on S_R. For example, the deficit may be quite large in a dry year, and small in the following year if it is particularly rainy. That doesn't mean that the root zone changed size over the span of one year. Only that a certain amount was detected.*

Reply:

This is indeed a delicate issue that we will explain in more detail. The reviewer is correct in stating that for any individual year the annual storage deficit is a mere lower bound of the necessary root zone storage capacity. For that reason we estimate the root zone storage capacity for higher dry spell return periods (i.e. 20 years for forest) as has been shown to be suitable for many different environments (e.g. Wang-Erlandsson et al., 2016). Evidence from optimality-based studies provide further evidence that vegetation does not dimension its root-systems much larger than that so that instead it can balance below-ground resource investment with above-ground growth that is needed in competition for light (e.g. Schymanski et al., 2008; Guswa, 2008).

*Comment:*

*There seems to be a lot of description concern about the impact of interception, followed by a decision to assign transpiration to be equal to all of ET. The paper could be simplified here.*

Reply:

Agreed. We will adjust that in the revised manuscript.

*Comment:*

*Eq. 3 effectively forces the system to be energy limited by scaling ET with PET. Is it not the case that water limitation ever occurs at any time of the year?*

Reply:

This is a very sharp observation. The reviewer is right, that here we have assumed energy limitation. We think this simplifying assumption is justified in the snow/glacier-melt dominated regime of the Alps, where early summer melt but also abundant summer rain coincide with energy supply, i.e. water and energy supply are in-phase, so that during times of largest atmospheric water demand also most water is available. This reduces the occurrence of water-limited conditions.

*Comment:*

*Eqs. 4 and 5 – why cast the deficit as negative? Confusing and a departure from most of the rest of the literature (e.g., Wang Erlandsson et al. 2016)*

Reply:

Agreed. We will adjust that in the revised manuscript.

*Comment:*

*Eq. 5 Is this equation correct? Or should it just be the minimum of the annual values – not their summation.*

Reply:

Agreed. This will be corrected.

**References:**

Bormann, H., Breuer, L., Gräff, T., & Huisman, J. A. (2007). Analysing the effects of soil properties changes associated with land use changes on the simulated water balance: A comparison of three hydrological catchment models for scenario analysis. *Ecological Modelling*, *209*(1), 29–40. https://doi.org/10.1016/j.ecolmodel.2007.07.004

Bouaziz, L. J. E., Aalbers, E. E., Weerts, A. H., Hegnauer, M., Buiteveld, H., Lammersen, R., Stam, J., Sprokkereef, E., Savenije, H. H. G., & Hrachowitz, M. (2022). Ecosystem adaptation to climate change: The sensitivity of hydrological predictions to time-dynamic model parameters. *Hydrology and Earth System Sciences*, *26*(5), 1295–1318. https://doi.org/10.5194/hess-26-1295-2022

Bulygina, N., Ballard, C., McIntyre, N., O'Donnell, G., & Wheater, H. (2012). Integrating different types of information into hydrological model parameter estimation: Application to ungauged catchments and land use scenario analysis. *Water Resources Research*, *48*(6). https://doi.org/10.1029/2011WR011207

Duethmann, D., Bloschl, G., & Parajka, J. (2020). Why does a conceptual hydrological model fail to correctly predict discharge changes in response to climate change? *Hydrology and Earth System Sciences*, *24*(7), 3493–3511. https://doi.org/10.5194/hess-24-3493-2020

Efstratiadis, A., & Koutsoyiannis, D. (2010). Une décennie d'approches de calage multi-objectifs en modélisation hydrologique: Une revue. In *Hydrological Sciences Journal* (Vol. 55, Issue 1, pp. 58–78). https://doi.org/10.1080/02626660903526292

Fatichi, S., Ivanov, V. Y., Paschalis, A., Peleg, N., Molnar, P., Rimkus, S., Kim, J., Burlando, P., & Caporali, E. (2016). Uncertainty partition challenges the predictability of vital details of climate change. *Earth's Future*, *4*(5), 240–251. https://doi.org/10.1002/2015EF000336

Gaur, S., Bandyopadhyay, A., & Singh, R. (2021). Projecting land use growth and associated impacts on hydrological balance through scenario-based modelling in the Subarnarekha basin, India. *Hydrological Sciences Journal*, *66*(14), 1997–2010. https://doi.org/10.1080/02626667.2021.1976408

Gentine, P., D'Odorico, P., Lintner, B. R., Sivandran, G., & Salvucci, G. (2012). Interdependence of climate, soil, and vegetation as constrained by the Budyko curve. *Geophysical Research Letters*, *39*(19), 2–7. https://doi.org/10.1029/2012GL053492

Guimberteau, M., Ciais, P., Pablo Boisier, J., Paula Dutra Aguiar, A., Biemans, H., De Deurwaerder, H., Galbraith, D., Kruijt, B., Langerwisch, F., Poveda, G., Rammig, A., Andres Rodriguez, D., Tejada, G., Thonicke, K., Von Randow, C., Randow, R., Zhang, K., & Verbeeck, H. (2017). Impacts of future deforestation and climate change on the hydrology of the Amazon Basin: A multi-model analysis with a new set of land-cover change scenarios. *Hydrology and Earth System Sciences*, *21*(3), 1455–1475. https://doi.org/10.5194/hess-21-1455-2017

Gupta, H. V., Wagener, T., & Liu, Y. (2008). Reconciling theory with observations: Elements of a diagnostic approach to model evaluation. *Hydrological Processes*, *22*(18), 3802–3813. https://doi.org/10.1002/hyp.6989

Guswa, A. J. (2008). The influence of climate on root depth: A carbon cost-benefit analysis. *Water Resources Research*, *44*(2), 1–11. https://doi.org/10.1029/2007WR006384

Hanus, S., Hrachowitz, M., Zekollari, H., Schoups, G., Vizcaino, M., & Kaitna, R. (2021). Future changes in annual, seasonal and monthly runoff signatures in contrasting Alpine catchments in Austria. *Hydrology and Earth System Sciences*, *25*(6), 3429–3453. https://doi.org/10.5194/hess-25-3429-2021

Huisman, J. A., Breuer, L., Bormann, H., Bronstert, A., Croke, B. F. W., Frede, H. G., Gräff, T., Hubrechts, L., Jakeman, A. J., Kite, G., Lanini, J., Leavesley, G., Lettenmaier, D. P., Lindström, G., Seibert, J., Sivapalan, M., Viney, N. R., & Willems, P. (2009). Assessing the impact of land use change on hydrology by ensemble modeling (LUCHEM) III: Scenario analysis. *Advances in Water Resources*, *32*(2), 159–170. https://doi.org/10.1016/j.advwatres.2008.06.009

Ibrahim, M., Coenders-Gerrits, M., Van Der Ent, R., & Hrachowitz, M. (n.d.). *Catchments do not strictly follow Budyko curves over multiple decades but deviations are minor and predictable*. https://doi.org/10.5194/hess-2024-120

Kirchner, J. W. (2006). Getting the right answers for the right reasons: Linking measurements, analyses, and models to advance the science of hydrology. *Water Resources Research*, *42*(3). https://doi.org/10.1029/2005WR004362

Merz, R., Parajka, J., & Blöschl, G. (2011). Time stability of catchment model parameters: Implications for climate impact analyses. *Water Resources Research*, *47*(2), 1–17. https://doi.org/10.1029/2010WR009505

Padulano, R., Costabile, P., Rianna, G., Costanzo, C., Mercogliano, P., & Del Giudice, G. (2022). *Comparing Different Modelling Strategies for the Estimation of Climate Change Effects on Urban Pluvial Flooding*. 5. https://doi.org/10.3390/environsciproc2022021005

Padulano, R., Rianna, G., Costabile, P., Costanzo, C., Del Giudice, G., & Mercogliano, P. (2021). Propagation of variability in climate projections within urban flood modelling: A multi-purpose impact analysis. *Journal of Hydrology*, *602*. https://doi.org/10.1016/j.jhydrol.2021.126756

Pechlivanidis, I. G., Arheimer, B., Donnelly, C., Hundecha, Y., Huang, S., Aich, V., Samaniego, L., Eisner, S., & Shi, P. (2017). Analysis of hydrological extremes at different hydro-climatic regimes under present and future conditions. *Climatic Change*, *141*(3), 467–481. https://doi.org/10.1007/s10584-016-1723-0

Porporato, A., Daly, E., & Rodriguez-Iturbe, I. (2004). Soil Water Balance and Ecosystem Response to Climate Change The terrestrial hydrologic cycle is an example of a manifold system whose understanding requires a synergistic use of. In *Am. Nat* (Vol. 164, Issue 5).

Reaver, N. G. F., Kaplan, D. A., Klammler, H., & Jawitz, J. W. (2022). Theoretical and empirical evidence against the Budyko catchment trajectory conjecture. *Hydrology and Earth System Sciences*, *26*(5), 1507–1525. https://doi.org/10.5194/hess-26-1507-2022

Samaniego, L., Kumar, R., & Attinger, S. (2010). Multiscale parameter regionalization of a grid-based hydrologic model at the mesoscale. *Water Resources Research*, *46*(5). https://doi.org/10.1029/2008WR007327

Schymanski, S. J., Roderick, M. L., Sivapalan, M., Hutley, L. B., & Beringer, J. (2008). A canopy-scale test of the optimal water-use hypothesis. *Plant, Cell and Environment*, *31*(1), 97–111. https://doi.org/10.1111/j.1365-3040.2007.01740.x

Stephens, C. M., Lall, U., Johnson, F. M., & Marshall, L. A. (2021). Landscape changes and their hydrologic effects: Interactions and feedbacks across scales. In *Earth-Science Reviews* (Vol. 212). Elsevier B.V. https://doi.org/10.1016/j.earscirev.2020.103466

Stephens, C. M., Marshall, L. A., & Johnson, F. M. (2019). Investigating strategies to improve hydrologic model performance in a changing climate. *Journal of Hydrology*, *579*. https://doi.org/10.1016/j.jhydrol.2019.124219

Tempel, N., Bouaziz, L., Taormina, R., Van Noppen, E., Stam, J., Sprokkereef, E., & Hrachowitz, M. (n.d.). *Vegetation Response to Climatic Variability: Implications for Root Zone Storage and Streamflow Predictions*. https://doi.org/10.5194/egusphere-2024-115

Tixeront, J.: *Prediction of streamflow (in French: Prévision des apports des cours d'eau),* IAHS publication n°63: General Assembly of Berkeley, IAHS, Gentbrugge, 118-126, http://hydrologie.org/redbooks/a063/063013.pdf, 1964.

Van Oorschot, F., Van Der Ent, R. J., Alessandri, A., & Hrachowitz, M. (2024). Influence of irrigation on root zone storage capacity estimation. *Hydrology and Earth System Sciences*, *28*(10), 2313–2328. https://doi.org/10.5194/hess-28-2313-2024

Van Oorschot, F., Van Der Ent, R. J., Hrachowitz, M., & Alessandri, A. (2021). Climate-controlled root zone parameters show potential to improve water flux simulations by land surface models. *Earth System Dynamics*, *12*(2), 725–743. https://doi.org/10.5194/esd-12-725-2021

Van Oorschot, F., Van Der Ent, R. J., Hrachowitz, M., Di Carlo, E., Catalano, F., Boussetta, S., Balsamo, G., & Alessandri, A. (2023). Interannual land cover and vegetation variability based on remote sensing data in the HTESSEL land surface model: Implementation and effects on simulated water dynamics. *Earth System Dynamics*, *14*(6), 1239–1259. https://doi.org/10.5194/esd-14-1239-2023

Vos, N. J. de, Rientjes, T. H. M., & Gupta, H. v. (2010). Diagnostic evaluation of conceptual rainfall-runoff models using temporal clustering. Hydrological Processes, 24(20), 2840–2850. https://doi.org/10.1002/hyp.7698

Wagener, T., Sivapalan, M., Troch, P., & Woods, R. (2007). Catchment Classification and Hydrologic Similarity. *Geography Compass*, *1*(4), 901–931. https://doi.org/10.1111/j.1749-8198.2007.00039.x

Wagner, W., Scipal, K., Pathe, C., Gerten, D., Lucht, W., & Rudolf, B. (2003). Evaluation of the agreement between the first global remotely sensed soil moisture data with model and precipitation data. *Journal of Geophysical Research: Atmospheres*, *108*(19). https://doi.org/10.1029/2003jd003663

Wang, S., Hrachowitz, M., & Schoups, G. (n.d.). *Multi-decadal fluctuations in root zone storage capacity through vegetation adaptation to hydro-climatic variability has minor effects on the hydrological response in the Neckar basin, Germany*. https://doi.org/10.5194/hess-2024-62

Wang-Erlandsson, L., Bastiaanssen, W. G. M., Gao, H., Jägermeyr, J., Senay, G. B., Van Dijk, A. I. J. M., Guerschman, J. P., Keys, P. W., Gordon, L. J., & Savenije, H. H. G. (2016). Global root zone storage capacity from satellite-based evaporation. *Hydrology and Earth System Sciences*, *20*(4), 1459–1481. https://doi.org/10.5194/hess-20-1459-2016

Reviewer #2

Upfront we would like to express our sincere gratitude to the reviewer for the time and effort invested in reading our manuscript. We highly appreciate the very constructive and insightful comments. Below we provide clarifications in detailed replies to all comments.

*Comment:*

*The thematic of parameter stationarity in climate impact studies is an important topic when aiming at generating suitable future water resources projections. The authors explore here a framework to ingrate possible evolution of root zone storage capacity ($S\_R$) in hydrological modelling experiments. The focus on alpine catchments in Austria and focus on impacts on streamflow response to changing soils. I like the topic and the way the authors present it. The findings represent a sample on the topic that might help future developments in this direction.*

*I list here below some thoughts and issues to be addressed.*

Reply:

We thank the reviewer for the positive feedback on our study and for recognizing the importance of parameter stationarity in climate impact studies, particularly for alpine catchments.

*Comment:*

*30 – 109: When accepting this review, I wondered if the authors would make a comment on Gao et al. (2023). I witnessed an interesting discussion on that between a hydrologist and a soil scientist. This was rather interesting, and I really wish the authors make their thoughts on it.*

Reply:

Thank you for pointing out the discussion surrounding Gao et al. (2023). The debate over whether soil characteristics—such as texture, porosity, and hydraulic conductivity—serve as primary determinants of water flow and moisture availability, or if, conversely, water movement is predominantly influenced by ecosystems and their responses to climatic factors, as suggested by Gao and colleagues, is indeed highly relevant in this context. We will include this in the introduction and discussion section of final manuscript to further underline the importance of ecosystem interactions for the partitioning of water, rather than soil properties alone.

*Comment:*

*69 – 84: The introduction on optimality is very well formulated. At line 534 you compare your findings to the ones presented in Speich et al. (2018). In the paper you cite optimality approaches are used and discussed. You could consider adding this study also in the introduction.*

Reply:

Agreed. We will include this in the final manuscript.

*Comment:*

*103 – 105: Staying with Speich et al research, they published in 2020 a study with transient evolution of root zone storage capacities in an alpine catchment according to different scenarios (Fig 9 in Speich et al., 2020). This might be relevant for you. Also, in Speich et al. simulations with and without dynamic S_R are presented and discussed.*

Reply:

Thank you for pointing out the research of Speich et al. (2020). Their study, investigating interactions between hydrological and forest dynamics under climate change scenarios in alpine catchments, is indeed relevant for our analysis. We will incorporate this reference the final manuscript.

*Comment:*

*113-115: The authors focus on alpine catchments with shallow soils. Is this more than an opportunistic choice stemming from the Hanus et al. (2021) study? I agree that there are the places where possibly most of soil genesis might occur, but is S_R not more dynamic in low-land areas? More in general, I see the trade-off in terms of "duplication" between this section 2 and section 2 in Hanus et al. (2021) as unproblematic.*

Reply:

The reviewer is correct in noting that our previous work (Hanus et al., 2021) investigated the effects of a changing climate on streamflow using the same model and study catchments in the Central Alps. However, the critical distinction in this study is the incorporation of potential changes in the role of vegetation, which was not considered in the earlier work. To allow comparability, the choice of catchments was thus guided by Hanus et al. (2021).

Soil genesis rates were not a decision criterion as previous work, including Gao et al. (2023), referred to by the reviewer above, but also many others (e.g. McCormick et al., 2021), demonstrate that subsurface water volumes accessible to vegetation are to the first order controlled by the extent of root systems rather than soil depth in many parts of the world.

We also acknowledge the reviewer's point that the catchments in this study are primarily cool and energy-limited. The $S_R$ parameter may indeed exhibit more dynamic behaviour in lowland areas, as shown by Bouaziz (2022) in the Meuse basin. This basin, located in NW-Europe, represents a hydro-climatically distinct region with an out-of-phase relationship between seasonal precipitation and energy signals. It experiences a winter rainfall regime, with peak flows in winter and low flows in summer.

In contrast, the alpine catchments in our current study feature a snow-dominated regime where the seasonal water supply, driven by snowmelt, glacier melt, and rainfall, is more in-phase with the energy signal, resulting in peak flows in summer and low flows in winter.

These contrasting characteristics, along with the vulnerability of snow-dominated alpine catchments to climate change, underscore our motivation for extending Bouaziz's methodology to this region. We will further emphasize this rationale in the revised manuscript.

*Comment:*

*Table 1: Is there any soil information to be included here?*

Reply:

We thank the reviewer for the for the suggestion. We will update Table 1 with underpinning soil information (e.g. soil texture fractions) in line with the SoilGrids250m dataset.

*Comment:*

*Figure 2: is a well-designed graphical abstract of the envisaged methodology. The indication of the colour schemes used is also very useful here. At the size presented in the submitted manuscript I don't see the reason for keeping the text at such small fonts. A couple of more points everywhere would ease the reader.*

Reply:

Agreed. We will update this in the final manuscript.

*Comment:*

*148 ff: Considering the large number of assumptions that are declared in the 5 steps of the methodology, wouldn't be useful to introduce some very basic benchmarks such as prescribed increase or decrease of S_R between the current and future time slice?*

Reply:

We thank the reviewer for the for this suggestion. We indeed agree with the reviewer that adding a sensitivity analysis would add value to our analysis. To show the sensitivity of the model towards the $S_R$ parameter, we will perform a targeted sensitivity analysis. Using the calibrated parameter sets and all other parameters held constant, we will test 15 distinct values for a catchment average $S_R$ (i.e., 50, 100, 150, 200, 250, 300, 350,...750), to introduce a high-contrast range in $S_R$ values. To maintain consistency, the $S_R$ values for the four landscape classes (i.e. vegetation types) will be scaled so that the catchment average $S_R$ remains unchanged. Using this "manipulated" parameter set we will rerun the model and assess the model's sensitivity to varying levels of $S_R$ under present conditions.

*Comment:*

*206 – 215: When you introduce the concept of supply and demand limited systems (also know as water and energy limited systems), wouldn't be beneficial to elaborate on the blue and green water paradox in the Alps? Cfr. Mastrotheodoros et al (2020).*

Reply:

We agree with the reviewer that addressing the Green-Blue Water Paradox would enhance the narrative by providing a tangible explanation of supply and demand-limited systems in the Alpine context. We will incorporate this in the final manuscript.

*Comment:*

*Figure 3: This Figure is presented in the Methodology but has also some results. As ω is fixed, we see no scatter in how the six basins respond to future climate. Have you done this exercise with past data (e.g. 1961-1990 vs 1991-2020) to see if such projections are realistic? (e.g. if ω is constant).*

Reply:

Thank you for the suggestion. Several recent studies, have demonstrated that, while ω cannot be assumed to be strictly constant over time, its temporal fluctuations with climatic variability are very minor in the vast majority of regions world-wide (e.g. Ibrahim et al., 2024; Tempel et al., 2024; Wang et al., 2024). Based on the results of these studies, we have therefore here not explicitly analyzed the validity of a fixed omega in a changing climate.. We will clarify that in the revised manuscript.

*Comment:*

*247 – 248: How did you partition for different vegetation types? Or did you just miss to refer to S2 also here?*

Reply:

We agree with the reviewer that this is not well explained in this section of the manuscript.

The results of previous studies (e.g., Wang-Erlandsson et al., 2016) suggest that different vegetation types adapt their root zones to bridge droughts of specific return periods. For instance, riparian vegetation, grasslands, and forests are assumed to adapt to withstand droughts with return periods of 2, 2, and 20 years, respectively. This approach allows us to distinguish between vegetation types based on their respective drought resilience.

As the Sr values from the water balance method represent the average catchment root zone storage capacity, we scale these values according to the proportional coverage of each vegetation type (i.e. HRU) in the catchment. We will clarify this in the final manuscript.

*Comment:*

*251: When you speak of observed and modelled past, I think you mean "past obtained from simulations with observed data" and "past obtained with data from GCM/RCM". If I am wrong, please explain me if I am correct, please state it clearly somewhere.*

Reply:

We thank the reviewer for pointing out this ambiguity. The reviewer is correct in his interpretation. When we refer to "observed past," we mean past conditions derived from simulations using observed climate data. On the other hand, "modelled past" refers to historical conditions generated using General Circulation Models (GCMs) or Regional Climate Models (RCMs). We will revise the manuscript to clearly define these terms to avoid any confusion.

*Comment:*

*Figure 4: The violins show especially in RCP85 a clustering of the outcomes in to two families. Does this relate to specific GMCs or RMCs?*

Reply:

We thank the reviewer for this valuable observation. It is indeed plausible that the observed clustering in outcomes is related to the variations among the RCMs used in the study. Although the analysis includes 14 RCMs, these models originate from only 5 distinct GCMs, which likely introduces a degree of homogeneity among RCMs derived from the same GCM. Consequently, the differences within RCMs generated from a single GCM are expected to be less pronounced than those between RCMs derived from different GCM families. This may be attributable to factors such as the GCMs' distinct underlying assumptions, resolution, or sensitivity to climate forcing. To investigate this hypothesis, we will identify the specific GCMs and RCMs associated with each cluster and analyse their characteristics to clarify the sources of clustering. We appreciate this suggestion and will incorporate any relevant findings into the revised manuscript.

*Comment:*

*Figure 5: I like This plot, which speaks a lot for plausibility of your approach, as you get decreasing S_R (and chance to survive) for forest in high altitude basins, while grassland seems to have more chance to survive. Is this an independent achievement from your procedure, or does this follow one of several constraints you defined? (lines 269-271)*

Reply:

We thank the reviewer for the insightful comment. We agree that this aspect was not thoroughly explained in the current manuscript, and we appreciate the opportunity to clarify further.

The observed decrease in $S_{R,clim}$ for forests, relative to the greater resilience observed in grasslands, is attributed to our modeling approach rather than to predefined constraints. The initial constraints were used solely for model calibration, affecting $S_{R,cal}$ parameter. After calibration, $S_{R,cal}$ was replaced by $S_{R,clim}$, which is derived using the water balance method. This method incorporates distinct return periods for different vegetation types and accounts for their areal distribution. Consequently, the results presented in Figure 5 are driven primarily by energy limitations within the catchments, which restrict root expansion, rather than by the initial constraints. We will clarify this in the revised manuscript.

Figure 7: *Did I miss a discussion on the cause of larger spread of "Modelled S_R_,clim" in the Pitztal? Glacier I guess?*

Reply:

We thank the reviewer for the insightful observation. From figure 7 it is evident that the Pitztal exhibits the largest spread in modeled streamflow simulations, both when using $S_{R,cal}$ and $S_{r,clim}$ parameters.

We agree that the larger spread observed is likely linked to the glacier presence in the Pitztal. In glacier-fed catchments, meltwater contributions have a strong influence on streamflow, and these contributions depend on factors such as temperature, seasonal fluctuations, and glacier dynamics. Modeling these complex dynamics introduces additional variability. In this study, glacial area changes are represented through linear interpolation of observed outlines from 1997 to 2006, with extrapolation to estimate glacier areas up to 2015. Melt is subsequently modeled using a degree-day method (Hanus, 2021). Including these additional parameters in the model inherently adds variability, contributing to a broader spread in both $S_{r,clim}$ and $S_{r,cal}$ simulations.

The difference in spread between the $S_{r,clim}$ and $S_{r,cal}$ model runs can be further explained by the significant narrowing down of the $S_R$ parameter range from calibration-based values to climate-based ones (Figure S5). The significantly lower $S_{R,clim}$ values represent a reduced capacity of the soil's to retain water. In glacier-dominated catchments, where snow and ice melt are primary sources of streamflow, this reduced retention capacity means that meltwater rapidly translates into runoff without sufficient soil storage. This lack of buffering amplifies fluctuations in streamflow, leading to larger extremes and increased variability between different model runs. Thus, the limited soil storage capacity in $S_{R,clim}$ heightens the model's sensitivity to climatic variations, causing greater discrepancies in streamflow simulations.

We agree that we have not highlighted this aspect sufficiently in the current manuscript and will revise this in the next iteration.

*Comment:*

*The whole analysis of signatures is solid and follows very closely the Hanus et al. (2021) pattern. I see here potential for shortening the paper. Instead of replicating all these analyses, I would prefer you explore the sensitity of S_R with more simple benchmarks as I suggest for lines 148 ff.*

Reply:

We acknowledge that some sections of the manuscript are overly lengthy and complex. We will condense these parts to enhance readability, while retaining key insights from Hanus et al. to maintain the manuscript's coherence and self-sufficiency.

Additionally, we will implement the sensitivity analysis, as described above, to further strengthen the study's robustness.

*Comment:*

*539 – 552*: I like this "Broader implication section". As far as the catchment selection part is concerned, it is a pity that the study only concentrated on the Hanus et al. basins, all of them being humid and energy limited. So this counts also as limitation

Reply:

Agreed. We will include this in the revised manuscript.

*Comment:*

*Final considerations: I like the organization and focus of this study. Having focus and analysed strongly connected with Hanus et al. (2021) is in first sight a good idea, but when looking at the results and analyses. I really considered a missed chance not including water limited catchments here. I fear that a follow-up study with addition of such basins would also be "jeopardized" by this "in between study". My honest request is here major revision with addition of ~6 water limited basins. Another option would the to keep it in this form but adding some simple S_R scenarios to test the sensitivity against your sophisticated approach.*

Reply:

We thank the reviewer for the insightful and constructive feedback! In principle, we agree that including water-limited catchments could enhance the analysis. However, we deliberately chose to perform this study in cool, energy-limited, to explore the method as proposed by Bouaziz et al. (2022) in a different climate. To address the reviewers' concern, we will add a sensitivity analysis using a high-contrast range of $S_R$ values to compare with our current results. This will allow us simultaneously address concerns regarding model sensitivity, while preserving the focus on the effect of future changes in root zone storage capacity on streamflow.

**References:**

Bouaziz, L. J. E., Aalbers, E. E., Weerts, A. H., Hegnauer, M., Buiteveld, H., Lammersen, R., Stam, J., Sprokkereef, E., Savenije, H. H. G., & Hrachowitz, M. (2022). Ecosystem adaptation to climate change: The sensitivity of hydrological predictions to time-dynamic model parameters. *Hydrology and Earth System Sciences*, *26*(5), 1295–1318. https://doi.org/10.5194/hess-26-1295-2022

Gao, H., Fenicia, F., & Savenije, H. H. G. (2023). HESS Opinions: Are soils overrated in hydrology? *Hydrology and Earth System Sciences*, *27*(14), 2607–2620. https://doi.org/10.5194/hess-27-2607-2023

Hanus, S., Hrachowitz, M., Zekollari, H., Schoups, G., Vizcaino, M., & Kaitna, R. (2021). Future changes in annual, seasonal and monthly runoff signatures in contrasting Alpine catchments in Austria. *Hydrology and Earth System Sciences*, *25*(6), 3429–3453. https://doi.org/10.5194/hess-25-3429-2021

Ibrahim, M., Coenders-Gerrits, M., Van Der Ent, R., & Hrachowitz, M. (n.d.). *Catchments do not strictly follow Budyko curves over multiple decades but deviations are minor and predictable*. https://doi.org/10.5194/hess-2024-120

Mastrotheodoros, T., Pappas, C., Molnar, P., Burlando, P., Manoli, G., Parajka, J., Rigon, R., Szeles, B., Bottazzi, M., Hadjidoukas, P., & Fatichi, S. (2020). More green and less blue water in the Alps during warmer summers. *Nature Climate Change*, *10*(2), 155–161. https://doi.org/10.1038/s41558-019-0676-5

McCormick, E. L., Dralle, D. N., Hahm, W. J., Tune, A. K., Schmidt, L. M., Chadwick, K. D., & Rempe, D. M. (2021). Widespread woody plant use of water stored in bedrock. *Nature*, *597*(7875), 225–229. https://doi.org/10.1038/s41586-021-03761-3

Speich, M. J. R., Lischke, H., & Zappa, M. (2018). Testing an optimality-based model of rooting zone water storage capacity in temperate forests. *Hydrology and Earth System Sciences*, *22*(7), 4097–4124. https://doi.org/10.5194/hess-22-4097-2018

Speich, M. J. R., Zappa, M., Scherstjanoi, M., & Lischke, H. (2020). FORests and HYdrology under Climate Change in Switzerland v1.0: A spatially distributed model combining hydrology and forest dynamics. *Geoscientific Model Development*, *13*(2), 537–564. https://doi.org/10.5194/gmd-13-537-2020

Tempel, N., Bouaziz, L., Taormina, R., Van Noppen, E., Stam, J., Sprokkereef, E., & Hrachowitz, M. (n.d.). *Vegetation Response to Climatic Variability: Implications for Root Zone Storage and Streamflow Predictions*. https://doi.org/10.5194/egusphere-2024-115

Wang, S., Hrachowitz, M., & Schoups, G. (n.d.). *Multi-decadal fluctuations in root zone storage capacity through vegetation adaptation to hydro-climatic variability has minor effects on the hydrological response in the Neckar basin, Germany*. https://doi.org/10.5194/hess-2024-62

Wang-Erlandsson, L., Bastiaanssen, W. G. M., Gao, H., Jägermeyr, J., Senay, G. B., Van Dijk, A. I. J. M., Guerschman, J. P., Keys, P. W., Gordon, L. J., & Savenije, H. H. G. (2016). Global root zone storage capacity from satellite-based evaporation. *Hydrology and Earth System Sciences*, *20*(4), 1459–1481. https://doi.org/10.5194/hess-20-1459-2016

**List of changes**

---

## Author Response (AR2)

Upfront we would like to express our sincere gratitude to the editor and reviewers for the time and effort invested in reading our revised manuscript. We greatly appreciate the thorough and precise review process, which has significantly contributed to improving the clarity of our work. Below, we address the final editorial comments.

*Comment:*

*Dear Authors,*

*Your revised manuscript has received favorable evaluations from the two reviewers and is close to final acceptance. As the final comments from the reviewers also indicate, the revised version remains as it is, but I suggest that the concluding section should give the reader the following additional information:*

*a) The selection of the catchments used in the study is centered more on legacy (i.e., previous works of most of the present authors) and less on adequacy.*

*b) Moreover, it should be emphasized that the present study refers to a limited geographic domain (i.e., energy-limited high alpine catchments), has an extremely large free parameter space, and a small parameter exploration space [changing root-zone soil-water storage (Sr)/vegetation distribution] to identify disparities in streamflows among the selected catchments.*

Reply:

We thank the editor and reviewers for the positive evaluation of our study and for the helpful suggestions. In response to the editorial comment, we have revised the concluding section to explicitly address the two points raised:

a) **Catchment selection based on legacy considerations**:
   We have clarified that the selection of catchments was guided primarily by prior work.

   *Line 625: Lastly, the study is confined to a narrow geographic scope of energy-limited alpine catchments, whose selection was guided by previous work*

b) **Emphasis on geographic and methodological constraints**:
   We have elaborated on the limitations related to geographic scope, the breadth of the free parameter space, and the narrow parameter exploration within the study.

   *Line 623: "Moreover, our analysis focuses exclusively on isolated changes in root zone storage parameter while keeping all other parameters constant, which limits the parameter exploration space and thereby the scope of streamflow disparities. Lastly, the study is confined to a narrow geographic scope of energy-limited alpine catchments, …"*

**List of changes**